# TLR2 regulates hair follicle cycle and regeneration via BMP signaling

Luyang Xiong[1†], Irina Zhevlakova[1†], Xiaoxia Z West[1], Detao Gao[2], Rakhilya Murtazina[1‡], Anthony Horak[3], J Mark Brown[3], Iuliia Molokotina[1], Eugene A Podrez[2], Tatiana V Byzova[1*]

[1]Department of Neurosciences, Lerner Research Institute, Cleveland Clinic, Cleveland, United States; [2]Department of Inflammation and Immunity, Lerner Research Institute, Cleveland Clinic, Cleveland, United States; [3]Department of Cardiovascular & Metabolic Sciences, Lerner Research Institute, Cleveland Clinic, Cleveland, United States

**\*For correspondence:**
byzovat@ccf.org

[†]These authors contributed equally to this work

**Present address:** [‡]Department of Biochemistry and Molecular Genetics, University of Illinois, Chicago, United States

**Abstract** The etiology of hair loss remains enigmatic, and current remedies remain inadequate. Transcriptome analysis of aging hair follicles uncovered changes in immune pathways, including Toll-like receptors (TLRs). Our findings demonstrate that the maintenance of hair follicle homeostasis and the regeneration capacity after damage depend on TLR2 in hair follicle stem cells (HFSCs). In healthy hair follicles, TLR2 is expressed in a cycle-dependent manner and governs HFSCs activation by countering inhibitory BMP signaling. Hair follicles in aging and obesity exhibit a decrease in both TLR2 and its endogenous ligand carboxyethylpyrrole (CEP), a metabolite of polyunsaturated fatty acids. Administration of CEP stimulates hair regeneration through a TLR2-dependent mechanism. These results establish a novel connection between TLR2-mediated innate immunity and HFSC activation, which is pivotal to hair follicle health and the prevention of hair loss and provide new avenues for therapeutic intervention.

## eLife assessment

Toll like receptor 2 (TLR2) signaling has traditionally been viewed a surface protein that induces innate immune responses and improves acquired immunity. Here, the authors suggest a different role for TLR2 in the hair cycle. By using a Cre reporter that is largely, but not solely active in hair follicle stem cells, the authors conditionally delete Tlr2 in mice and report that BMP signaling is sustained and hair cycle entry is delayed. Delving further, the authors identify CEP (2-$\omega$-carboxyethyl pyrrole) as an endogenous ligand of TLR2 in hair follicle stem cell regulation. Although a role for TLR2 signaling in hair follicle stem cells is potentially novel and **important**, the reviewers remain in consensus that evidence presented in two significant areas continues to be **incomplete**: (1) where TLR2 and CEP are expressed and how specific is their expression to the hair follicle stem cells; (2) whether as the authors suggest, TLR2 functions by regulating BMP signaling in the stem cell niche of the hair follicle.

## Introduction

Hair follicles (HFs) represent one of the best examples of mini-organs with the ability to regenerate throughout life, which, in turn, relies on the proliferation and differentiation of HF stem cells (HFSCs) within hair bulge (*Fuchs and Blau, 2020*; *Sakamoto et al., 2021*). The cyclic renewal of HFs is orchestrated by the interplay between inhibitory and stimulatory signals (*Plikus et al., 2011*). Despite the immune privileged status of HFs, they have a unique microbiome and immune system, including

resident macrophages and other immune cells (*Bertolini et al., 2020*; *Fuchs and Blau, 2020*; *Paus et al., 2003*). Components of the HF immune system have been implicated in regulating the HF cycle and its regeneration (*Di Domizio et al., 2020*; *Rahmani et al., 2020*). Given their exposure to pathogens, HFs are equipped with innate immune receptors, particularly Toll-like receptors (TLRs), which detect and respond to pathogens by stimulating the secretion of defensins (*Di Domizio et al., 2020*; *Selleri et al., 2007*).

TLRs play a key role in recognizing and responding to either pathogen- or damage-associated molecular patterns, mediating the cytokine response. However, the role of TLRs extends beyond this function, as they have been shown to directly promote tissue regeneration and homeostasis in multiple tissues, particularly in stem and progenitor cells. TLRs regulate hematopoietic and intestinal stem cell renewal, proliferation, and apoptosis (*Nagai et al., 2006*; *Tomchuck et al., 2008*). The role of innate immune responses in the tissue-healing benefits of stem cell therapy has been clearly demonstrated (*Vagnozzi et al., 2020*). Moreover, TLR activation is a critical component of the reprogramming or transdifferentiation of adult cells into pluripotency (*Lee et al., 2012*), emphasizing the close coordination between innate immunity, cell transformation, and regeneration.

Multiple reports connect altered HFs' immunity to hair loss, including a breakdown of immune privilege in alopecia areata (*Rahmani et al., 2020*). Likewise, androgen, which is tightly linked to TLR activation, was shown to influence the innate immunity of HFs in androgenic alopecia (*Sawaya, 2012*). The decline of innate immunity processes due to aging or conditions like obesity is widely recognized and these conditions are causatively associated with hair thinning and loss (*Andersen et al., 2016*; *Ghanemi et al., 2020*; *Palmer and Kirkland, 2016*; *Shaw et al., 2013*). Alopecia patients often have higher body weight index and weight compared to healthy individuals (*Bakry et al., 2014*). Increased body weight index is linked to more significant hair loss severity in adults (*Goette and Odom, 1976*) and a higher prevalence of hair disorders in children and adolescents (*Mirmirani and Carpenter, 2014*). Mouse models support these findings, showing that activation of innate immunity through pathogen signals might lead to alopecia (*Shin et al., 2018*) and that high-fat diets inducing obesity cause hair thinning through HFSC depletion (*Morinaga et al., 2021*).

Our previous studies have shown that activating endothelial TLR2 by endogenous ligands such as CEP (a product of PUFA oxidation) promotes wound healing and tumor angiogenesis (*West et al., 2010*). Deletion of TLR2 from endothelial cells reduces tumor size by diminishing its vasculature (*McCoy et al., 2021*). In wounded skin, endothelial TLR2 is crucial for tissue regeneration through increased proangiogenic cytokine secretion (*Xiong et al., 2022b*). Although PUFAs have been shown to benefit hair growth by extending the anagen phase and promoting cell proliferation and hair shaft elongation (*Munkhbayar et al., 2016*), the role of innate immunity and in particular, TLR2 in the HF cycle remains unknown.

Using animal models and human cell lines, we show a new function of TLR2 in the HF cycle in homeostasis and HF regeneration in injury. Furthermore, we demonstrate that an endogenously produced PUFA metabolite CEP serves as a TLR2 ligand in the hair bulge, promoting hair regeneration and growth through TLR2. In conditions associated with hair loss, i.e., aging and obesity, both TLR2 and its ligand are substantially depleted in HFs.

## Results
### TLR2 in HF declines due to aging and obesity

To assess whether and how aging affects HF innate immunity, we analyzed available RNA sequencing data of mouse HFSCs (*Doles et al., 2012*). Pathway analysis revealed that innate and adaptive immunity, as well as TLR signaling, were among the top dysregulated pathways (*Figure 1A*). Notch, JAK-STAT, TGF-β, and Wnt, and other pathways essential for HF regeneration were also altered by aging (*Figure 1A*). Notably, the level of *Tlr2* mRNA in HFSCs of old mice was 2-fold lower compared to young mice (*Keyes et al., 2013*). In addition, TLR2 at the protein level was substantially lower in 13-month-old mice compared to 2-month-old mice (*Figure 1B and C*).

At the same time, the normal hair cycle is marked by an increase in *Tlr2* mRNA levels prior to HFSC activation during telogen, according to the analysis of existing RNA microarray data (*Figure 1D*; *Greco et al., 2009*). *Tlr2* mRNA levels are highest among other *Tlr*s, with *Tlr1* and *Tlr4* mRNA showing the opposite pattern. TLR6 may act as a co-receptor for TLR2 as its expression pattern is similar. While

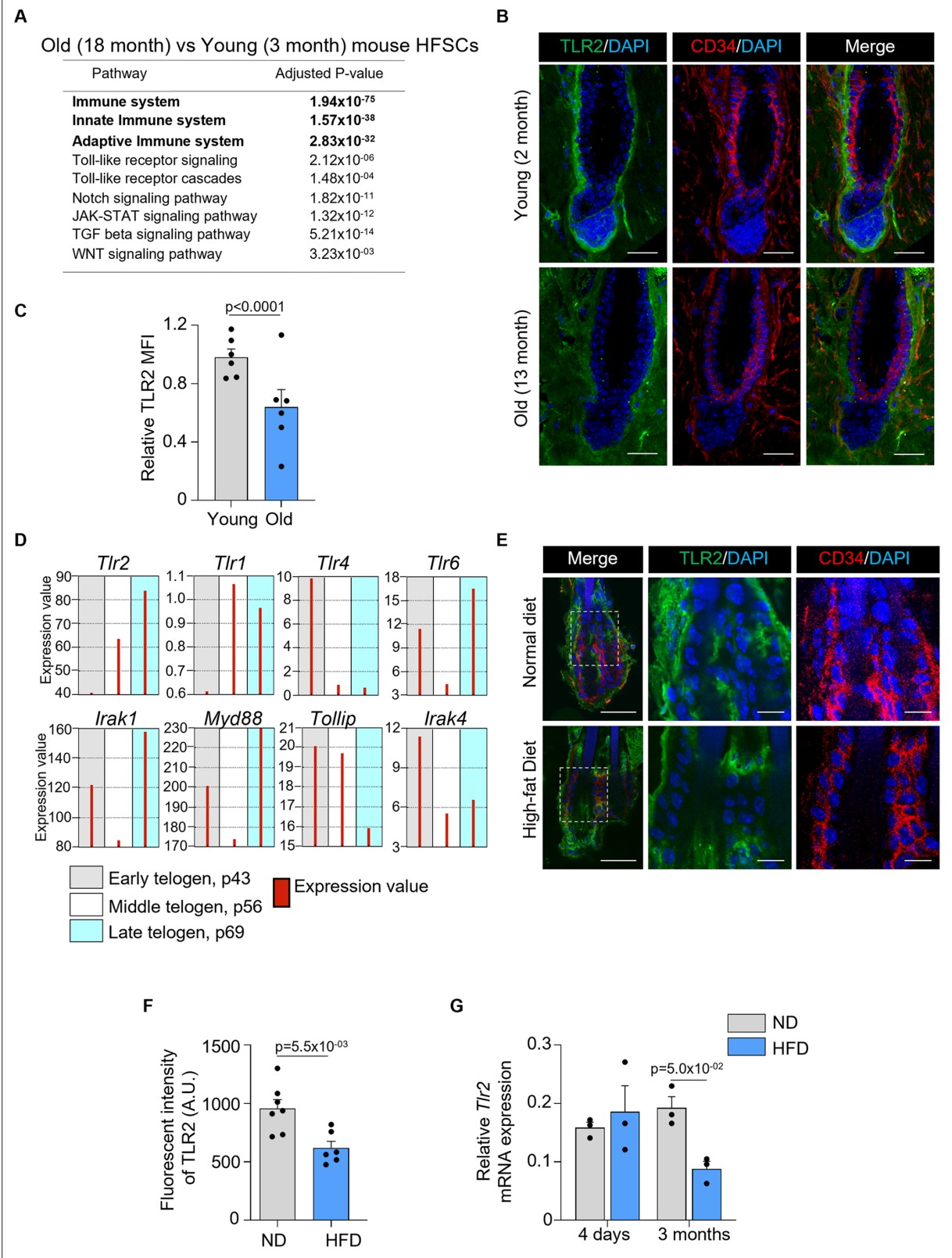

**Figure 1.** Hair follicle stem cells (HFSCs) downregulate TLR2 in response to stress like a high-fat diet and aging. (**A**) Dysregulated pathways in old vs young mouse HFSCs. The top pathways are labeled in bold. (**B**) Representative confocal images of telogen hair follicles from young and old mice immunostained for TLR2 and CD34 demonstrate decreased TLR2 intensity in HFSC (CD34-positive) of old mice. Scale bars are 50 μm. The middle and right panels show a magnified view of the boxed area. Scale bars are 20 μm. (**C**) Quantification of TLR2 fluorescent intensity in images from B showing

*Figure 1 continued*

significantly lower TLR2 expression in HFSCs from the old mice. N=6 for each group. (**D**) GEO2R analysis of published RNA data from sorted follicle populations in the second telogen to anagen transition demonstrates the increased level of *Tlr2* mRNA accompanied by the activation of Toll-like receptors (TLRs) signaling downstream. (**E**) Representative confocal images showing TLR2 expression in hair follicles from mice fed with a normal diet (ND) or high-fat diet (HFD). CD34 is an HFSC marker. Scale bars are 50 μm. Magnified images demonstrate decreased TLR2 intensity in HFSC (CD34-positive) of mice after HFD. Scale bars are 20 μm. (**F**) Quantification of TLR2 fluorescent intensity in images from E showing significantly lower TLR2 expression in HFSCs from HFD-fed mice. N=7 and 6 for ND and HFD groups, respectively. AU, arbitrary unit. (**G**) *Tlr2* mRNA expression in HFSCs from mice fed with ND or HFD for 4 days or 3 months. Data regenerated from published RNA sequencing dataset GSE131958. N=3 for each group. All bar graphs are mean ± s.e.m. Non-parametric Mann-Whitney test (**C, G**) or unpaired two-tailed t-test (**F**) was used to determine statistical difference. A p-value ≤ 0.05 was considered to be statistically significant.

the downstream TLR2 signaling molecules, *Irak1* and *Myd88*, are also upregulated, mirroring the *Tlr2*, IRAK1 inhibitor *Tollip* is suppressed. Together, the results suggest the critical regulatory role of the entire TLR2/TLR6 pathway in the HF cycle.

High-fat diet-induced obesity causes hair thinning and subsequent loss (*Morinaga et al., 2021*). In our model, a high-fat diet causes a nearly 2-fold decline in TLR2 protein level in HFSCs, compared to normal diet-fed mice (*Figure 1E and F*). Further, RNA sequencing data reveal that 3 months of a high-fat diet is sufficient to reduce *Tlr2* levels in HFSCs by more than 2-fold compared to control mice (*Figure 1G*; *Morinaga et al., 2021*). Thus, our results and analyses of existing datasets demonstrate that conditions causatively associated with hair thinning and loss, such as aging and obesity, result in a dramatic depletion of TLR2 in HFSCs suggesting a possible regulatory role for TLR2 in HFs.

## TLR2 is upregulated during HFSC activation

The expression of TLR2 during a normal hair cycle was assessed using a previously characterized TLR2-GFP reporter mouse (*Figure 2*), one of the best tools for the analysis of TLRs in vivo (*Price et al., 2018*). The correlation between the reporter and TLR2 protein expression was confirmed (*Figure 2—figure supplement 1A and B*). The HF cycle was verified by H&E staining (*Figure 2—figure supplement 1C*). During telogen, a dormant stage for HFSCs, TLR2 was found in the bulge, secondary hair germ (sHG), dermal papilla (DP), and outer root sheath (ORS) (*Figure 2A*). During regenerative anagen, TLR2 expression was detected in the DP and all sHG-derived progenitor cells (*Figure 2B and C*) and quiescent bulge stem cells (*Figure 2D and E*). All cells derived from bulge stem cells (ORS) and sHG (hair shaft and inner root sheath [IRS]) were positive for TLR2 (*Figure 2C*). In catagen, TLR2 was abundant in the new bulge and sHG formed from ORS cells (*Hsu et al., 2011*; *Figure 2F and G*). While TLR2 was expressed in early sHG lineage (including IRS) (*Figure 2C*), it was absent in mature (*Figure 2D and E*) and regressing (*Figure 2F*) IRS. This shows that TLR2 is abundant in stem cells but declines upon differentiation. The second telogen's old and new bulges expressed TLR2 (*Figure 2H*). TLR2 was present in sHG and DP during the second telogen (*Figure 2H*) and increased in the late (competent) telogen compared to the early (refractory) telogen (*Figure 2—figure supplement 1D and E*). The highest level of TLR2 occurred during active anagen compared to quiescent telogen and catagen (*Figure 2I*). Quantitative polymerase chain reaction (qPCR) revealed that the *Tlr2* mRNA level in HFSCs was 5- and 2.3-fold higher in anagen than in telogen and catagen, respectively (*Figure 2J*). Notably, analysis of existing RNA sequencing data using FACS-sorted cells (*Lorz et al., 2010*) confirmed that TLR2 expression was significantly higher in HFSCs than in epidermal or non-stem cells (*Figure 2K*). Thus, TLR2 is enriched in HFSCs, and its expression increases during activation.

## Deletion of Tlr2 in HFSCs delays anagen onset in the normal hair cycle

To address the role of TLR2 in the hair cycle, we generated an HFSC-specific inducible *Tlr2* knockout (KO) mouse line (TLR2[HFSC-KO]) and deleted *Tlr2* during the first postnatal telogen (*Xiong et al., 2022b*). In TLR2[HFSC-KO] mice, the telogen phase was substantially prolonged compared to control (*Tlr2[lox/lox]*) mice, as summarized in the schematic (*Figure 3A*). Melanogenesis and anagen onset are tightly coupled (*Müller-Röver et al., 2001*). Thus, the pink skin color at P21 marks the first postnatal telogen. The onset of anagen in control mice was indicated by the change in skin color to gray or black at P26 in control mice. TLR2[HFSC-KO] mice at this age did not enter anagen, as evidenced by the delayed darkening of their skin (*Figure 3A–C*). This was further confirmed by skin section analysis at P21, P26, and P35 (*Figure 3D*). At P26, control mice displayed pigmented anagen HFs

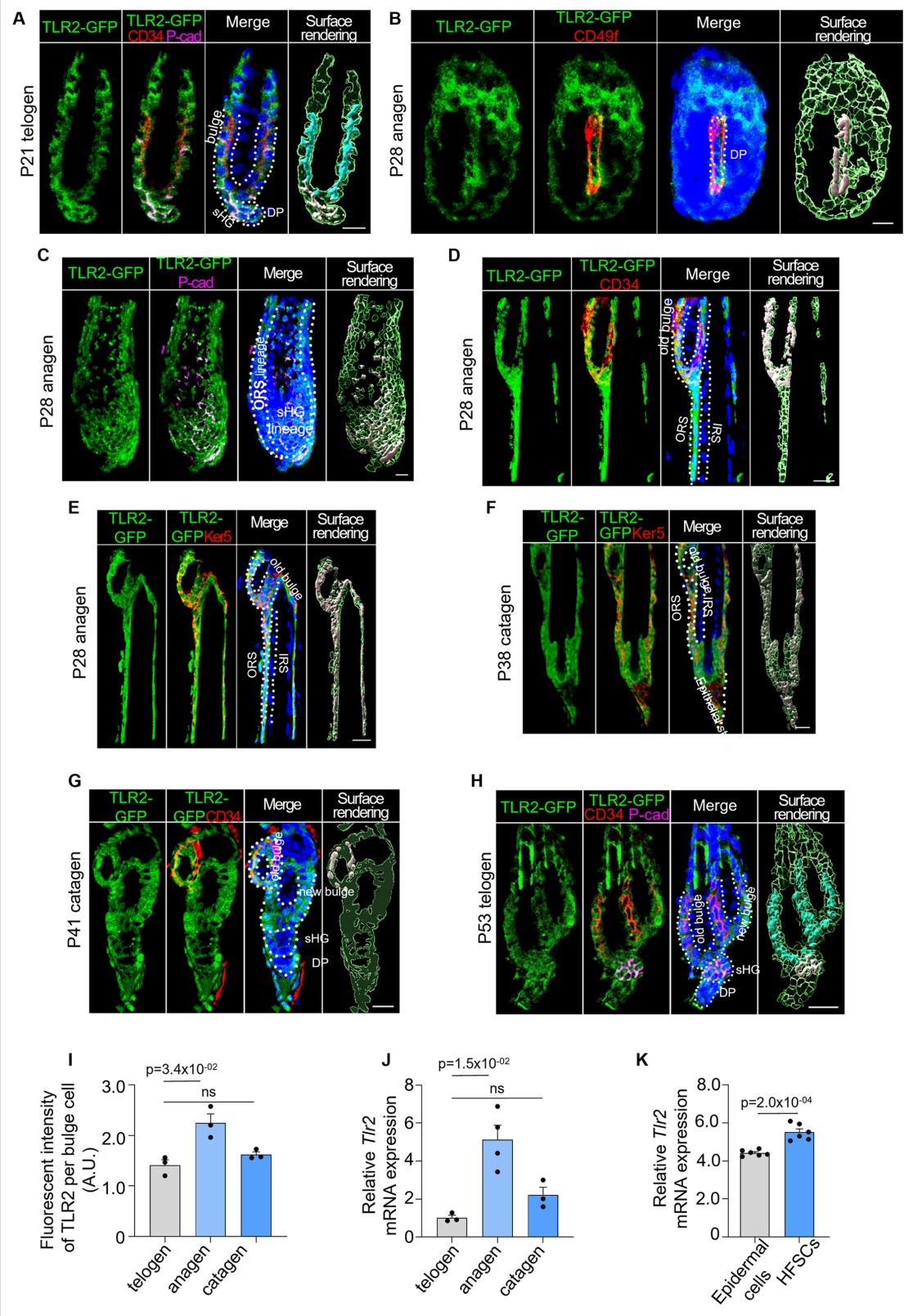

**Figure 2.** TLR2 is enriched in hair follicle stem cells (HFSCs) and is upregulated during HFSC activation. TLR2-GFP reporter mouse skin sections were immunostained with anti-GFP to assess TLR2 expression in the hair follicles. (**A**) Representative confocal images of P21 first telogen hair follicle immunostained for TLR2-GFP, CD34 (bulge stem cells), P-cad (secondary hair germ [sHG]), and DAPI (nuclei). The green color in the surface rendering panel represents TLR2 expression, and other surfaces show co-localization between TLR2 and specific markers. TLR2 is present in bulge, sHG, and

*Figure 2 continued on next page*

*Figure 2 continued*

dermal papilla (DP) cells. P represents postnatal days. Scale bar is 10 μm. (**B**) TLR2-GFP in P28 anagen was co-immunostained with CD49f of basement membrane outlining the DP. Scale bar is 10 μm. (**C**) TLR2 is co-localized to the sHG lineage (P-cad+ layers), DP, and outer root sheath (ORS) lineage. Scale bar is 20 μm. (**D**) TLR2-GFP in P28 anagen was co-immunostained with CD34 in old bulge (**D**) and Ker5 in ORS (**E**) revealing TLR2 localization to the old bulge, ORS, but not inner root sheath (IRS). Scale bars are 20 μm. (**F**) Co-immunostaining of TLR2-GFP in P38 catagen hair follicle with Ker5 in ORS lineage cells showing co-localization of TLR2 with ORS and bulge. Scale bar is 20 μm. (**G**) P41 late catagen hair follicle immunostained for TLR2 and CD34 showing co-localization of TLR2 to the old bulge, new bulge, sHG, and DP. Scale bar is 20 μm. (**H**) P53 second telogen hair follicle immunostained for TLR2, CD34, and P-cad reveals co-localization of TLR2 to the bulge, sHG, and DP. Scale bar is 20 μm. (**I**) Quantification of TLR2 fluorescent intensity in bulge cells at different phases showing TLR2 upregulation in anagen. N=3 for each group. (**J**) Quantitative polymerase chain reaction (qPCR) analysis of *Tlr2* mRNA expression in FACS-purified mouse HFSCs in anagen, telogen, and catagen. N=3 or 4 per group. (**K**) qPCR analysis of *Tlr2* mRNA expression in mouse epidermal cells and FACS-purified HFSCs showed significantly higher *Tlr2* expression in HFSCs compared with raw epidermal cells. N=6 mice per group. All bar graphs are mean ± s.e.m. Two-tailed unpaired t-test (**K**) or Kruskal-Wallis test with Dunn's post hoc test (**I, J**) was used to determine statistical difference. A p-value ≤ 0.05 was considered to be statistically significant.

The online version of this article includes the following figure supplement(s) for figure 2:

**Figure supplement 1.** TLR2/GFP correlation in immunostaining.

with enlarged bulbs located deeply in the hypodermis, while nearly all follicles of TLR2[HFSC-KO] mice were ~5-fold shorter and remained in the dermis on top of adipose tissue, a characteristic of telogen. On day P35, we observe partial entrance into anagen in TLR2[HFSC-KO] skin while the skin color of TLR2[HFSC-KO] mice remains pink (*Figure 3D–F*). HFSCs were activated as early as at P24 in control mice based on positive Ki67 staining in sHG and bulge region (*Figure 3G*), while most cells in TLR2[HFSC-KO] sHG (*Figure 3H*) and bulge (*Figure 3I*) remained quiescent. At P25, control mice exhibited a large cluster of P-cad+ cells encapsulating DP within the transformed sHG (*Figure 3K*), whereas the sHG of TLR2[HFSC-KO] mice remained small and inactive (*Figure 3J and K* and *Figure 3—figure supplement 1*). Despite the substantial delay in anagen onset, the morphology of HFs and expression of established HFSC markers, including Ker15, CD34, and Sox9, were normal in TLR2[HFSC-KO] mice (*Figure 3—figure supplement 2*). Thus, TLR2 in HFSCs is essential for HFSC activation and progression of the hair cycle.

## TLR2 regulates HFSC activation by interacting with BMP signaling pathway

The relationship between Wnt and BMP signaling is central to the cyclic growth of HFs (*Plikus et al., 2008*). Anagen initiation is triggered by Wnt/β-catenin activation, while BMP signaling suppresses HFSC activation and its reduction is necessary for HFSC activation. Indeed, during the early (refractory) phase of the second telogen, HFSCs exhibit elevated BMP signaling as evidenced by high levels of BMP7 protein and pSMAD1/5/9, downstream targets of BMP signaling, compared to the late (competent) phase (*Figure 4A–D*). In contrast, *Bmp7* and its effectors, *Id1* and *Id2*, are decreased during the late telogen based on our analysis of existing RNA microarray (*Greco et al., 2009*; *Figure 4—figure supplement 1A*).

To assess the possible connection between the TLR2 signaling and BMP pathway in human cells, we activated TLR2 and BMP signaling in human epidermal keratinocytes (NHEK) using a canonical TLR2 agonist (Pam3CSK4) and BMP4, respectively. As anticipated, BMP4 promoted the phosphorylation of its downstream target SMAD1/5/9. However, simultaneous co-activation of TLR2 diminished BMP4 signaling (*Figure 4—figure supplement 1B and C*).

Likewise, stimulation of human HFSC with canonical TLR2 agonist Pam3CSK4 promoted cell proliferation by 1.5-fold compared to controls. Notably, this effect was diminished in the presence of a TLR2-blocking antibody (*Figure 4—figure supplement 1D and E*). These results reveal that TLR2 activation on human HFSC augments their proliferation.

To gain a deeper understanding of TLR2's role in HFSC activation, we profiled the transcriptome of HFSCs lacking *Tlr2* expression (*Figure 4—figure supplement 1F*). The results showed that *Tlr2* deletion dysregulated 486 genes, many of which were involved in both the hair cycle and innate immunity. The most affected pathways included innate immunity response and TLR2 signaling, with its downstream target NF-kappaB (*Figure 4—figure supplement 1G*). This profile somewhat resembles changes observed in aging models (*Figure 1A*).

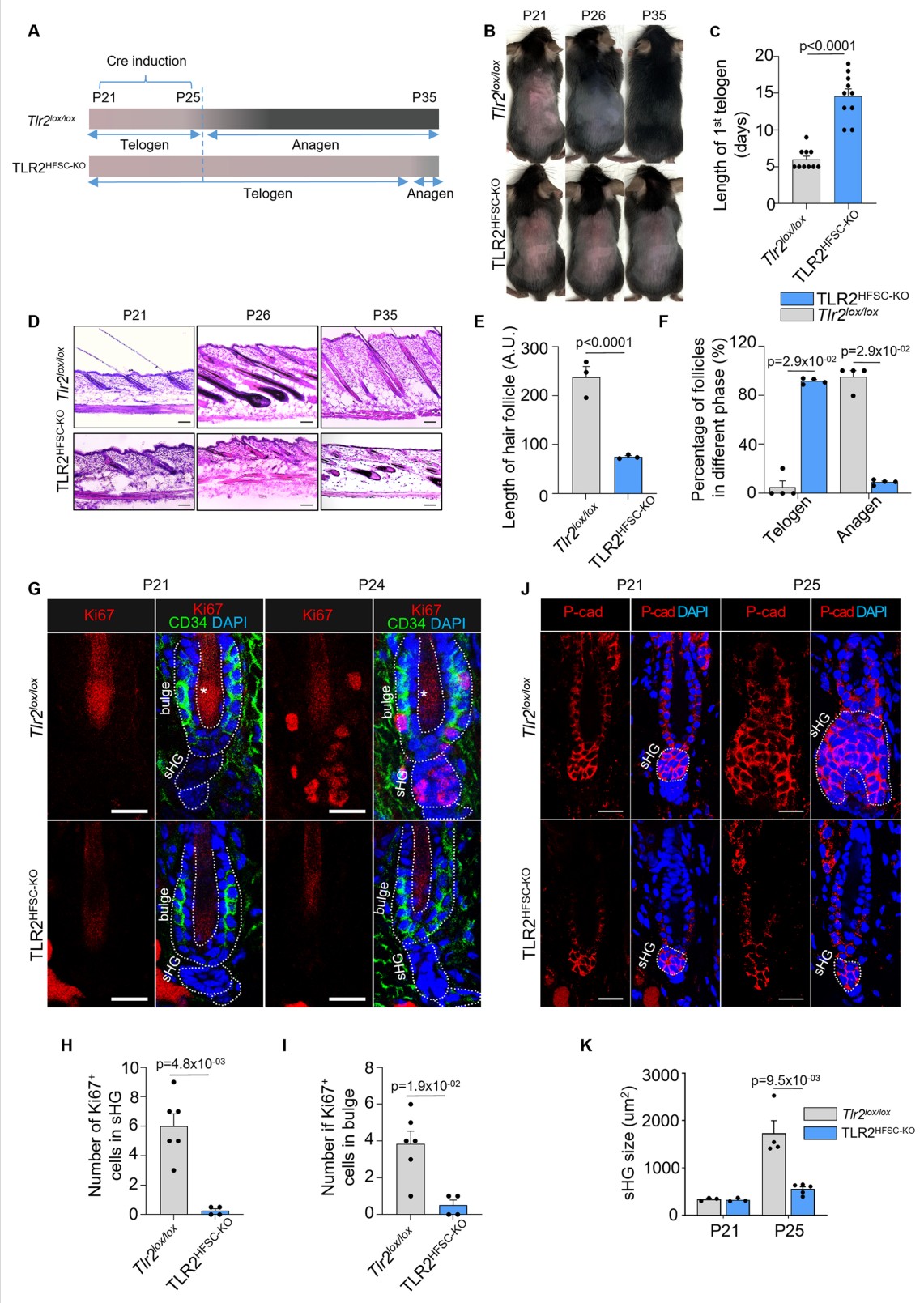

**Figure 3.** Deletion of TLR2 in hair follicle stem cells delays anagen onset. (**A**) Schematic of RU486-mediated Cre induction and dorsal skin pigmentation change (gradient bars) in *Tlr2^lox/lox* and TLR2^HFSC-KO mice. (**B**) Representative images of shaved *Tlr2^lox/lox* and TLR2^HFSC-KO mice showing different phases of the hair cycle. The *Tlr2^lox/lox* mouse transitions from telogen (pink skin) to anagen (gray/black skin) at P26 and a full hair coat is developed by P35. The TLR2^HFSC-KO mouse exhibits a prolonged telogen (**P21–P30–P35**). Representative images of at least 10 mice in each group. (**C**) Bar graph showing

*Figure 3 continued on next page*

*Figure 3 continued*

the length of first postnatal telogen starting from P21 measured by skin color change from B. N=10 per group. (**D**) Representative H&E staining of dorsal skin at indicated time points showing prolonged telogen in TLR2$^{HFSC-KO}$ mice. Scale bars are 50 μm. (**E**) The length of hair follicles at P26 from images in D. 50 hair follicles from three mice per group were used for quantification. (**F**) Percentages of telogen or anagen hair follicles at P26 from D. N=4 mice per group. (**G**) Representative confocal images of P21 and P24 first telogen hair follicles from *Tlr2$^{lox/lox}$* and TLR2$^{HFSC-KO}$ mice immunostained for CD34, Ki67, and DAPI. Stars label the hair shaft. Scale bars are 20 μm. (**H, I**) Quantification of images in G shows a diminished number of Ki67$^+$ cells in secondary hair germ (sHG) (**H**) and in CD34$^+$ bulge (**I**) in TLR2$^{HFSC-KO}$ mice compared to *Tlr2$^{lox/lox}$* at P24. N=6 and 4 mice for *Tlr2$^{lox/lox}$* and TLR2$^{HFSC-KO}$ group, respectively. (**J**) Representative confocal images of P21 and P25 dorsal skin sections from *Tlr2$^{lox/lox}$* and TLR2$^{HFSC-KO}$ mice immunostained for P-cad and DAPI showing changes in the size of sHG. Scale bars are 20 μm. (**K**) Quantification of sHG size in panel K shows enlarged sHG in *Tlr2$^{lox/lox}$* mice compared with TLR2$^{HFSC-KO}$ mice. N=4 mice for P25 *Tlr2$^{lox/lox}$*, and N=5 mice for TLR2$^{HFSC-KO}$. Statistical significance was determined using a non-parametric Mann-Whitney test. All data are mean ± s.e.m. A p-value ≤ 0.05 was considered to be statistically significant.

The online version of this article includes the following figure supplement(s) for figure 3:

**Figure supplement 1.** Confocal images of hair follicles immunostained for P-cad secondary hair germ (sHG) enlargement and elongation at anagen onset.

**Figure supplement 2.** Bulge stem cell marker expression in TLR2$^{HFSC-KO}$ mouse.

## BMP pathway is altered in TLR2 KO HFSCs

Since TLR2 suppresses BMP signaling and promotes HFSC proliferation, we assessed whether the delayed anagen in TLR2$^{HFSC-KO}$ mice might be associated with the BMP pathway. qPCR analysis reveals that several key components of the BMP pathway were dysregulated in HFSCs lacking *Tlr2* (*Figure 4E*). Among those, the most notable changes were observed for *Bmp7*, which was upregulated by ~4-fold in TLR2-null HFSCs compared to controls (*Figure 4E*). This was substantiated by co-staining of tissue sections for BMP7 and CD34, which demonstrated an ~2-fold increase in BMP7 on HFSCs of TLR2$^{HFSC-KO}$ mice as compared to the control (*Figure 4F and G*). Activation of BMP signaling was assessed by pSMAD1/5/9 positive staining in HF. Quantification revealed an ~15-fold higher level of pSMAD1/5/9 in TLR2-null mice (TLR2$^{KO}$) as compared to controls (wild type [WT]) (*Figure 4—figure supplement 1H and I*). The significant increase in BMP signaling observed was attributed to the absence of TLR2 in HFSCs. This was evidenced by a comparable 14-fold increase in BMP signaling in follicles of TLR2$^{HFSC-KO}$ mice compared to control during the first postnatal telogen phase, thereby ensuring the preservation of follicles in the dormant telogen stage (as shown in *Figure 4H and I*). Simultaneously, the Wnt signaling and β-catenin stabilization within HFSCs, known to trigger their activation (*Deschene et al., 2014*), remained unchanged between control and TLR2$^{HFSC-KO}$ mice (as shown in *Figure 4—figure supplement 1J*).

## BMP antagonist rescues defects caused by the lack of HFSCs TLR2

To demonstrate that an altered BMP pathway is, indeed, responsible for the phenotype of TLR2 KO in HFSCs, we utilized intradermal injection of noggin, a well-known inhibitor of BMP signaling (*Botchkarev et al., 2001*; *Botchkarev et al., 1999*), to block the upregulated BMP signaling in TLR2$^{HFSC-KO}$ mice. As a result, noggin injection diminished activation of BMP signaling by >10-fold in TLR2$^{HFSC-KO}$ mice as assessed by pSMAD1/5/9 staining of HF (*Figure 4J and K*). Moreover, noggin promoted activation of TLR2$^{HFSC-KO}$ HFs while the HFs in BSA-treated TLR2$^{HFSC-KO}$ mice remained quiescent. Noggin treatment of TLR2$^{HFSC-KO}$ mice dramatically upregulated cell proliferation within sHG as evidenced by Ki67$^+$ cells (*Figure 4L and M*), promoting ~2.5-fold increase in activated follicles (*Figure 4N and O*), thereby contributing to nearly 2-fold larger sHG (*Figure 4P and Q*) as compared to BSA-treated TLR2$^{HFSC-KO}$ mice. Thus, curbing suppressive BMP signaling in TLR2$^{HFSC-KO}$ mice can reactivate their HFs, demonstrating a causative connection between TLR2 and BMP pathways in the hair cycle.

## HFSC TLR2 governs hair regeneration upon injury

High expression of TLR2 and its critical role in HFSC activation during the hair cycle prompted us to test the role of HFSCs TLR2 in an injury model where cells are more likely to be exposed to TLR2 ligands. First, we compared TLR2 levels in HFs in wounded and healthy skin using TLR2-GFP reporter mouse (*Figure 5A*). In healthy skin, HFSCs upregulated TLR2 during their transition from middle to late telogen (day 5 to day 10) (*Figure 5B*, upper panels, and gray bars in *Figure 5C*), consistent with RNA sequencing results (*Greco et al., 2009*). This increase in TLR2 precedes HFSCs activation

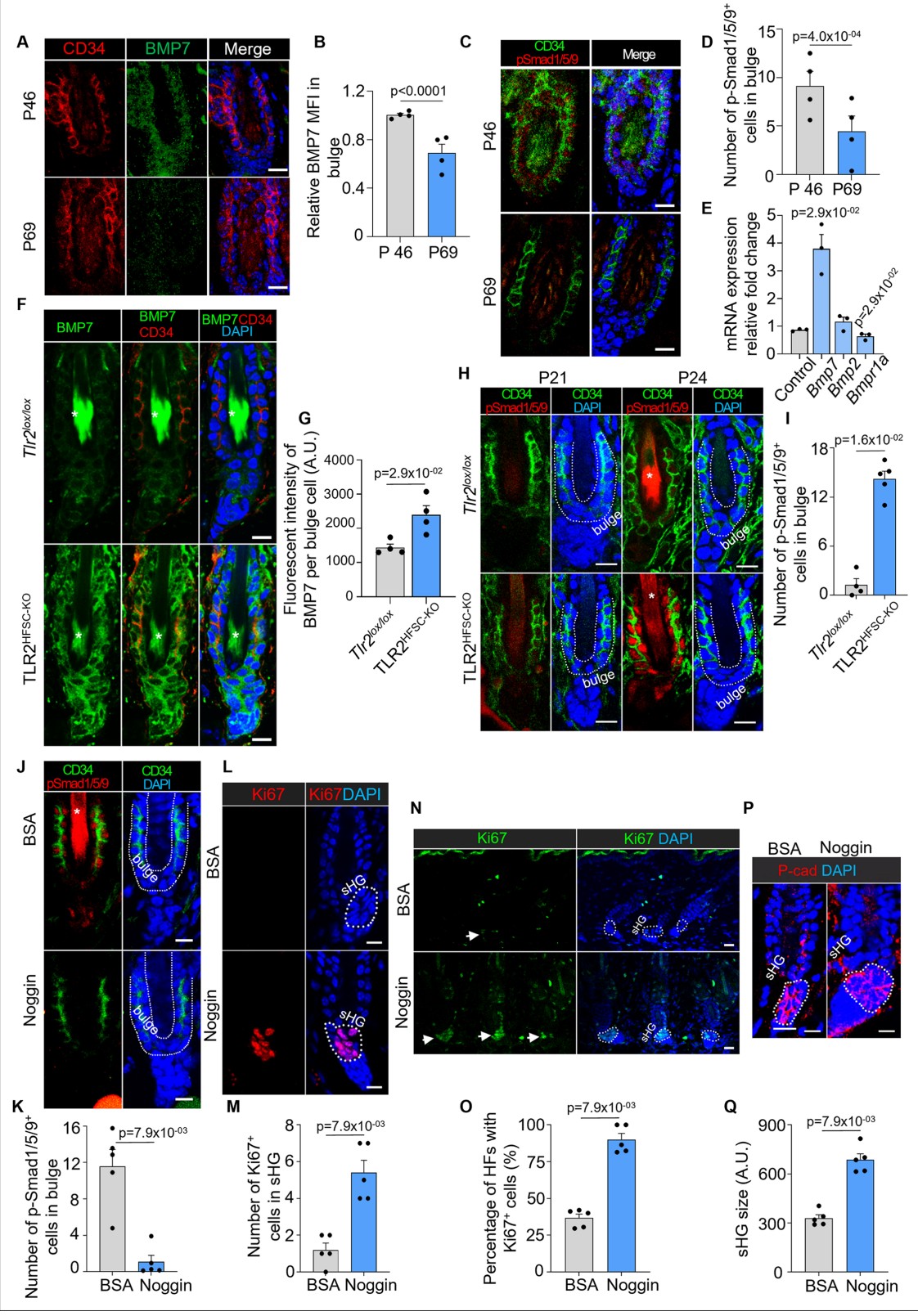

**Figure 4.** TLR2 interacts with BMP pathway to regulate the hair cycle. (**A**) Representative confocal images of BMP7 staining in hair follicles of dorsal skin in early (P46) and late (P69) second telogen. Scale bars are 10 μm. (**B**) Quantification of BMP7 fluorescent intensity from A showing diminished BMP7 expression during the second telogen from the early to late phases. N=4 per group. (**C**) Representative confocal images of pSMAD1/5/9 staining in hair follicles of dorsal skin in early (P46) and late (P69) second telogen. Scale bars are 10 μm. (**D**) Quantification of pSMAD1/5/9+ positive cells in CD34+

*Figure 4 continued on next page*

*Figure 4 continued*

bulge stem cells demonstrates a decrease of pSmad1/5/9 expression in late telogen. N=4 per group. (**E**) Quantitative polymerase chain reaction (qPCR) analysis reveals dysregulation of BMP singling molecules in hair follicle stem cells (HFSCs) lacking *Tlr2*. N=4 mice for control and *Bmp2*, N=3 mice for *Bmp7* and *Bmpr1a*. (**F**) Representative confocal images of BMP7 staining in hair follicles from *Tlr2^lox/lox^* or TLR2^HFSC-KO^ mice. Scale bars are 10 µm. Stars label hair shaft. (**G**) Quantification of BMP7 fluorescent intensity from F showing higher BMP7 expression in TLR2^HFSC-KO^ mice. N=4 per group. (**H**) P21 and P24 dorsal skin sections from *Tlr2^lox/lox^* and TLR2^HFSC-KO^ mice immunostained for CD34, pSmad1/5/9, and DAPI. Scale bars are 10 µm. (**I**) Quantification of pSmad1/5/9^+^ cells in CD34^+^ bulge stem cells in P24 dorsal skin from H. N=4 and 5 for *Tlr2^lox/lox^* and TLR2^HFSC-KO^ respectively. (**J**) Representative confocal images of dorsal skin sections from TLR2^HFSC-KO^ mice treated with BSA or noggin immunostained for CD34, pSmad1/5/9, and DAPI. Star labels the hair shaft. Scale bars are 10 µm. (**K**) Quantification of pSmad1/5/9^+^ cells in CD34^+^ bulge stem cells from images in J. N=5 per group. (**L**) Immunostaining for Ki67 and DAPI in dorsal skin sections from TLR2^HFSC-KO^ mice treated with BSA or noggin. Scale bars are 10 µm. (**M**) Quantification of images in L showing an increase in Ki67^+^ cells in secondary hair germ (sHG) of noggin-treated compared to BSA-treated TLR2^HFSC-KO^ dorsal skin. N=5 per group. (**N**) Representative confocal images of Ki67 and DAPI immunostaining of dorsal skin sections from TLR2^HFSC-KO^ mice treated with BSA or noggin. Arrows point to hair follicles with Ki67^+^ cells in the sHG. Scale bars are 20 µm. (**O**) Quantification of images in N showing percentages of hair follicles with Ki67^+^ cells in sHG. N=5 per group. (**P**) BSA- or noggin-treated TLR2^HFSC-KO^ mouse dorsal skin immunostained for P-cad and DAPI. The dashed line outlines the sHG. Scale bars are 10 µm. (**Q**) Bar graph showing significantly larger sHG in noggin-treated TLR2^HFSC-KO^ mice. N=5 per group. Mann-Whitney test was used to determine the statistical significance. All data are mean ± s.e.m. A p-value ≤ 0.05 was considered to be statistically significant.

The online version of this article includes the following source data and figure supplement(s) for figure 4:

**Figure supplement 1.** TLR2-BMP axis in hair follicle cells.

**Figure supplement 1—source data 1.** Uncropped WB gels.

during the normal cycle. However, in wound HFSCs, TLR2 was upregulated immediately after an injury resulting in 1.5-fold higher expression compared to normal unwounded skin (*Figure 5B and C*).

Hair regeneration after injury represents a substantial part of the healing process (*Abbasi and Biernaskie, 2019*; *Chen et al., 2015*; *Wang et al., 2017*). We next assessed the role of TLR2 using age- and gender-matched WT and TLR2^KO^ mice. The lack of *Tlr2* visibly impaired hair regeneration after wound healing (*Figure 5D*). On day 14 post-injury, HFs in WT mice entered precocious anagen judged by a spot of pigmented skin, which, on day 21, developed into a black hair patch (*Figure 5D*). In contrast, the follicles of TLR2^KO^ mice remained quiescent lacking regenerated HFs around wounds even after 21 days post-injury (*Figure 5D and E*). At this point, the pigmented skin area in WT was ~9-fold larger than in TLR2^KO^ mice. Skin flaps showed substantial pigmentation and growing hair bulbs around wounds in WT, indicative of active anagen. In contrast, the TLR2^KO^ skin flap was devoid of pigmentation, consistent with telogen (inner skin flap in *Figure 5D*). Ki67 staining confirmed an increase in HFs' activation in WT but not in TLR2^KO^ skin (*Figure 5F and G*). The resulting density of regenerated HFs based on Ker17 staining in WT was 2-fold higher than in TLR2^KO^ mice (*Figure 5H and I*). Most importantly, this effect was dependent on TLR2, specifically on HFSCs, since TLR2^HFSC-KO^ mice exhibited a similar phenotype with a dramatic reduction in pigmentation and hair growth compared to control mice (*Figure 5J and K*). The upregulation of pSmad1/5/9 in TLR2^HFSC-KO^ wounds compared to controls demonstrates that similar to the HF cycle scenario, increased BMP signaling might contribute to diminished HF regeneration (*Figure 5L and M*). Thus, the TLR2-BMP axis in HFSCs governs HF regeneration after injury.

## Endogenous ligand promotes hair regeneration via TLR2 on HFSCs

One of the most important endogenous ligands for TLR2 is CEP, which is a naturally occurring product of PUFA oxidation shown to be accumulated during inflammation and wound healing (*West et al., 2010*; *Xiong et al., 2022a*). Healthy tissues are typically devoid of this product, which is mainly associated with inflammation and pathologies (*Yakubenko et al., 2018*). However, in contrast to other tissues, healthy HFs exhibited high levels of CEP accumulation (*Figure 6A*). During anagen, CEP is present within the proximal part of the follicle, while in telogen the entire follicle is encased by this PUFA metabolite (*Figure 6B–E*). Generation of CEP from PUFA is directly aided by myeloperoxidase (MPO) (*Xiong et al., 2022a*; *Yakubenko et al., 2018*). MPO is present in abundance in sebaceous glands, possibly as a part of immune defense (*Figure 6—figure supplement 1A*). Even more surprising, in contrast to other organs and tissues, CEP in HFs is substantially depleted with age (*Figure 6F and G*), and this decline coincides with the reduction in the regenerative potential of HFs. This is likely due to a decreased level of MPO during aging (*Figure 6—figure supplement 1B and C*).

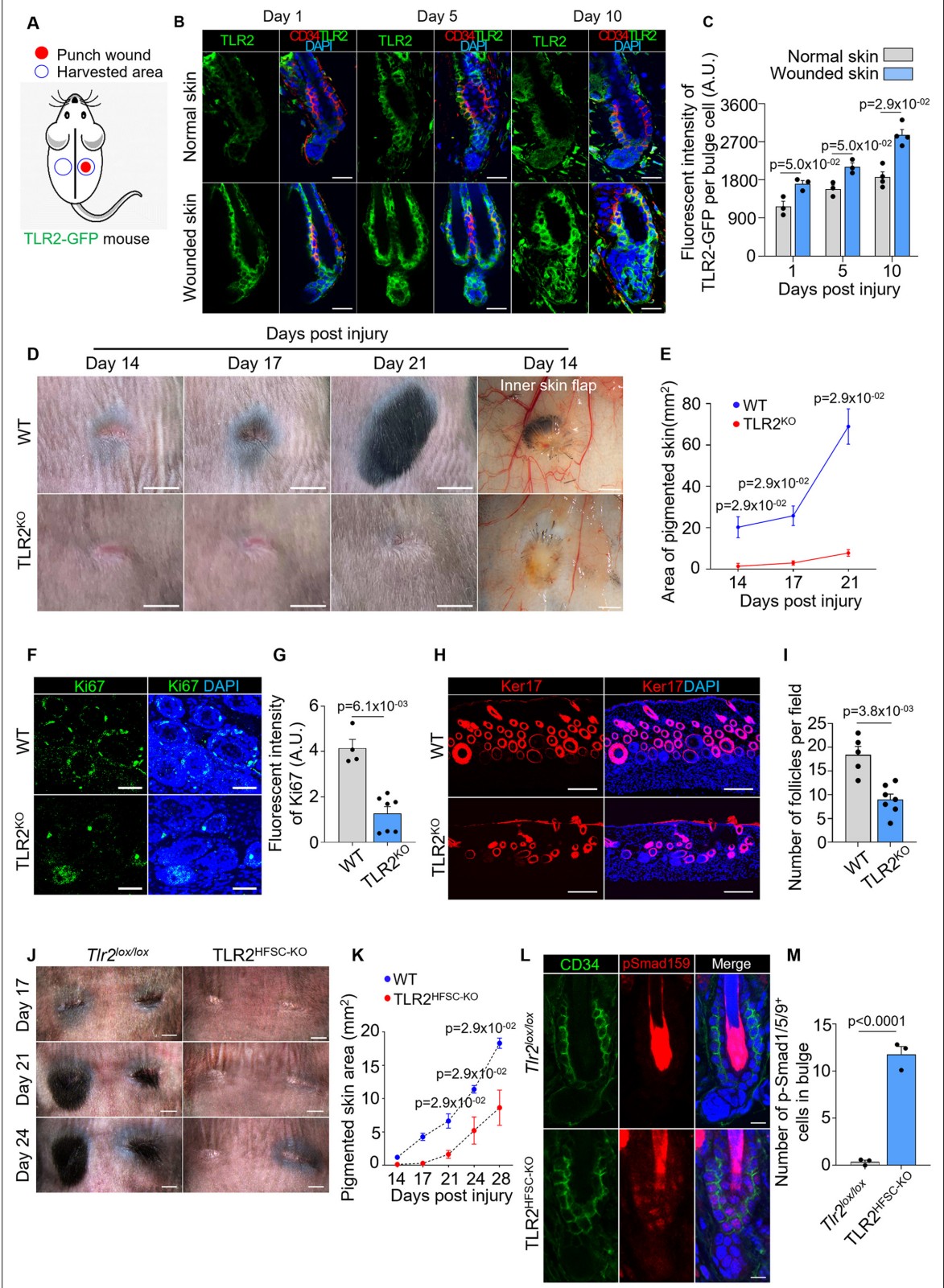

**Figure 5.** Hair follicle stem cell (HFSC) TLR2 is crucial for wound-induced hair follicle regeneration. (**A**) Schematic of wound healing assay using TLR2-GFP reporter mouse. Full-thickness wounds on the dorsal skin of TLR2-GFP mice were created. Normal unwounded skin and the skin adjacent to the wound were harvested at different time points. (**B**) Representative confocal images of normal and wounded skin from TLR2-GFP mice at different time points post-injury immunostained for TLR2-GFP and CD34. Scale bars are 20 µm. (**C**) Quantification of TLR2 fluorescent intensity per bulge cell in hair

*Figure 5 continued on next page*

*Figure 5 continued*

follicles from B shows increased TLR2 level in hair follicles from wounded skin as compared to normal skin. N=3 for day 1 and day 5, and N=4 for day 10 per group. (**D**) Representative photographs showing hair regeneration on the dorsal skin and inner skin flaps at indicated time post-injury in wild-type (WT) and TLR2 global knockout (TLR2$^{KO}$) mice. Diminished hair growth around the wound is apparent in TLR2$^{KO}$ skin from day 14 through 21 post-injury compared to WT skin. The inner skin flaps from TLR2$^{KO}$ at day 14 post-injury show an absence of pigmented hair bulbs and skin pigmentation. Scale bars are 5 mm for dorsal skin and 1 mm for inner skin flaps. (**E**) Quantification of the pigmented dorsal skin area around the wound from images in D shows diminished pigmentation in TLR2$^{KO}$ skin compared with WT skin at all time points post-injury. N=4 per group. (**F**) Representative confocal images of skin adjacent to wound immunostained for Ki67 and DAPI. Scale bars are 50 μm. (**G**) Bar graph showing diminished Ki67 fluorescent intensity in the skin adjacent to wound in TLR2$^{KO}$ mouse compared to WT mouse from images in F. N=4 and 7 for WT and TLR2$^{KO}$ respectively. (**H**) Representative confocal images of skin adjacent to wound immunostained for Ker17 and DAPI. Scale bars are 100 μm. (**I**) Quantification of hair follicle numbers from images in H reveals a significant decrease in regenerated hair follicles in TLR2$^{KO}$ skin compared with WT skin. N=5 and 7 for WT and TLR2$^{KO}$ respectively. (**J**) Representative photographs showing a lack of hair regeneration and skin pigmentation around the wound on the dorsal skin of TLR2$^{HFSC-KO}$ mice compared with *Tlr2$^{lox/lox}$* mice on day 17, day 21, and day 24 post-injury. Scale bars are 2 mm. (**K**) Quantification of pigmented skin area around the wound during 14–28 days post-injury showing significantly smaller pigmented skin area in TLR2$^{HFSC-KO}$ mice compared with *Tlr2$^{lox/lox}$* mice. N=4 per group. (**L**) Representative confocal images of wounded skin from *Tlr2$^{lox/lox}$* and TLR2$^{HFSC-KO}$ mice stained for CD34 and pSmad1/5/9. Scale bars are 10 μm. (**M**) Quantification of images from L showing more pSmad1/5/9$^+$ cells in TLR2$^{HFSC-KO}$ wounded skin. N=3 per group. Mann-Whitney test was used to determine the statistical significance. All data are mean ± s.e.m. A p-value ≤ 0.05 was considered to be statistically significant.

The connection between CEP levels and hair thinning and loss in aging prompted us to test whether exogenous CEP can activate TLR2 in HFSCs and stimulate their proliferation. Our in vitro experiments revealed that CEP increases the proliferation of human HFSC in a TLR2-dependent manner since the blockade of TLR2 abrogates the CEP effect (***Figure 6—figure supplement 1D and E***). In another model, CEP promotes cell proliferation of human hair follicle dermal papilla cells (HFDPCs) by ~2-fold compared to control (***Figure 6—figure supplement 1F***).

Next, we show that CEP promotes hair regeneration in injury in a TLR2-dependent manner. CEP administration promoted HF regeneration in WT wounds. However, it was ineffective in global TLR2$^{KO}$ mice (***Figure 6—figure supplement 1H***). CEP promoted a 55% increase in the number of HF and cell proliferation in WT wounds, and at the same time, there was no effect in TLR2$^{KO}$ wounds (***Figure 6— figure supplement 1I,J***).

To ensure independence from immune cells, WT and TLR2$^{KO}$ mice were irradiated and transplanted with WT bone marrow prior to wounding. Applying CEP on wounds in WT/WT chimeras promoted cell proliferation, thereby dramatically increasing the density of HFs (***Figure 6H and I***, ***Figure 6—figure supplement 1G***). At the same time, CEP was not effective in TLR2$^{KO}$/WT mice (***Figure 6H and I***, ***Figure 6—figure supplement 1G***), demonstrating the TLR2-dependent mechanism.

These CEP effects were mediated by TLR2 on HFSCs. In control mice, CEP effectively initiated regeneration of HFs around the wound (***Figure 6J***), resulting in ~3-fold higher density of HF (***Figure 6K and L***) and dramatic acceleration of cell proliferation by >10-fold (***Figure 6M and N***) as compared to TLR2$^{HFSC-KO}$ mice where CEP was mainly ineffective. A similar stimulatory effect of CEP was observed in a primary keratinocyte culture (***Oshimori and Fuchs, 2012***). CEP dramatically promoted WT but not TLR2$^{KO}$ keratinocyte proliferation (***Figure 6O and P***). CEP was an effective stimulator of TLR2 signaling as judged by augmented *Nfkb2*, *Il1b*, and *Il6* expression in HFSCs upon treatment with CEP (***Figure 6Q***). Consistent with the key role of BMP signaling in TLR2-dependent HF regeneration, CEP treatment suppressed inhibitory *Bmp7* expression by ~2.5-fold (***Figure 6R***), demonstrating that endogenous and natural TLR2 ligand can counteract an inhibitory effect of BMP7 to stimulate HFSCs' activation (***Figure 6S***).

## Discussion

The main findings of this study are as follows: (1) Expression of TLR2 in HFSCs is decreased with aging and in a mouse model of obesity. (2) In young and healthy animals, TLR2 expression in HFs is cycle-dependent, with the highest expression in HFSCs during the initiation of the anagen phase. (3) The absence of TLR2 in HFSCs prolongs the resting phase of the hair cycle and significantly delays hair regeneration after injury. (4) TLR2 regulates the hair cycle primarily by inhibiting BMP signaling in HFSCs. (5) HFs continuously produce a metabolite of PUFAs, which acts as an endogenous TLR2 ligand and promotes hair growth through TLR2 activation in HFSCs. Besides reduced TLR2, aging

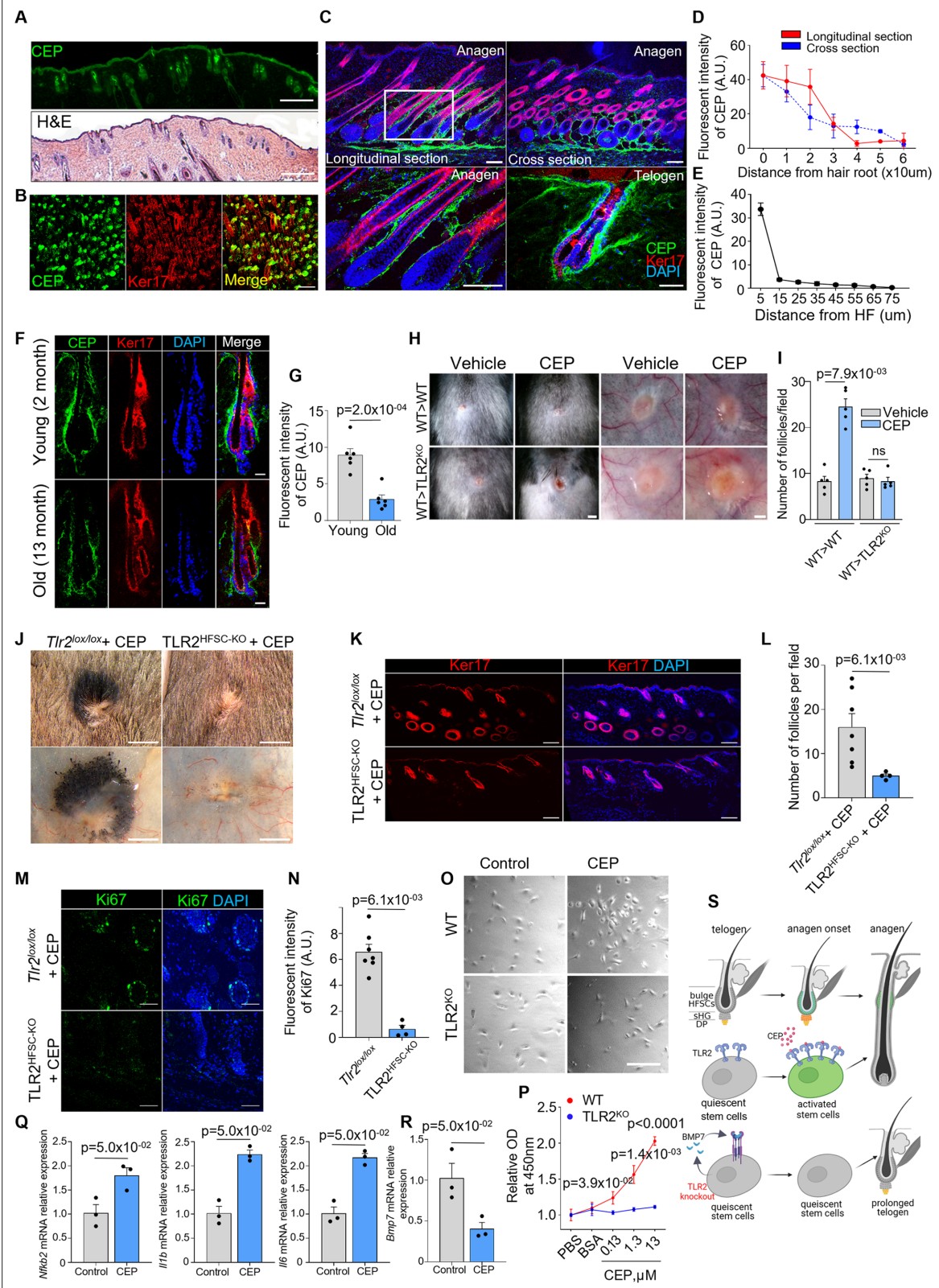

**Figure 6.** Oxidation-dependent TLR2 ligand carboxyethylpyrrole (CEP) is present in hair follicles and promotes hair regeneration via hair follicle stem cell (HFSC) TLR2. (**A**) Representative images of H&E and CEP immunostaining of consecutive skin sections from wild-type (WT) anagen mouse. Scale bars are 1 mm. (**B**) Representative confocal images of P5 WT whole-mount skin immunostained for CEP and Ker17. The merged image shows the co-localization of CEP to anagen hair follicles (Ker17+). Scale bar is 200 μm. (**C**) Longitudinal and cross-sections of anagen and telogen hair follicles from

*Figure 6 continued on next page*

*Figure 6 continued*

WT mice immunostained for CEP and Ker17. The lower left panel shows a magnified view of the boxed area. Scale bars are 100 μm for anagen, 50 μm for telogen. (**D**) Quantification of CEP fluorescent intensity at a different distance from the root of anagen hair follicles in longitudinal and cross-sections immunostaining images in images from C. A gradual decrease in CEP levels is observed from the proximal to the distal part of anagen hair follicles. N=50 follicles from 3 mice per group. (**E**) Line chart showing a sharp decrease of CEP fluorescent intensity with the distance from HF in telogen (from the lower right panel in C). N=10 follicles from 3 mice per group. (**F**) Representative confocal images of telogen hair follicles from young and old mice immunostained for CEP and Ker17. Scale bars are 20 μm. (**G**) Quantification of CEP fluorescent intensity from images in F. N=6 mice per group. (**H**) Representative photographs of dorsal skin (two left panels) and inner skin flaps (two right panels) from WT and TLR2$^{KO}$ mice after irradiation and bone marrow transplantation of WT bone marrow demonstrate an increased number of pigmented hair bulbs and skin pigmentation around wounds in CEP-treated wounds compared to control in WT mice with no differences in TLR2$^{KO}$ transplanted with WT bone marrow. Scale bars are 1 mm for the dorsal skin and 500 μm for the inner skin flap. (**I**) Quantitative results from H show an increased density of hair follicles upon CEP application around wounds of WT>WT transplanted mice with no changes in WT>TLR2$^{KO}$ mice. N=5 for each group. (**J**) Representative photographs of dorsal skin (upper panels) and inner skin flaps (lower panels) from *Tlr2$^{lox/lox}$* and TLR2$^{HFSC-KO}$ mice treated with CEP show a lack of pigmentation around TLR2$^{HFSC-KO}$ wounds compared with *Tlr2$^{lox/lox}$* wounds treated with CEP. The inner skin flap of TLR2$^{HFSC-KO}$ mice demonstrates an absence of pigmented hair bulbs after the CEP treatment. Scale bars are 3 mm. (**K**) Representative confocal images of skin adjacent to wound immunostained for Ker17. Scale bars are 100 μm. (**L**) Quantification of hair follicle numbers in images from K reveals a significant decrease in regenerated hair follicles in TLR2$^{HFSC-KO}$ skin compared with *Tlr2$^{lox/lox}$* skin. N=7 for *Tlr2$^{lox/lox}$*. N=4 for TLR2$^{HFSC-KO}$. (**M**) Representative confocal images of skin adjacent to wound immunostained for Ki67. Scale bars are 50 μm. (**N**) Bar graph showing Ki67 fluorescent intensity in the skin adjacent to wound from images in M. N=7 for *Tlr2$^{lox/lox}$*. N=4 for TLR2$^{HFSC-KO}$. (**O**) Representative microphotographs of primary keratinocytes isolated from WT or TLR2$^{KO}$ mouse skin co-cultured with CEP or control (PBS or BSA). Representative images from at least three independent assays are shown. Scale bar 50 μm. (**P**) Cell proliferation of primary keratinocytes in O indicates increased proliferation by CEP in WT but not in TLR2$^{KO}$ keratinocytes. N=3 independent experiments. (**Q**) Quantitative polymerase chain reaction (qPCR) analyses of *Nfkb2, Il1b,* and *Il6* mRNA levels in FACS-purified mouse HFSCs treated with BSA control or CEP. N=3 per group. (**R**) qPCR analyses of *Bmp7* mRNA levels in FACS-purified mouse HFSCs treated with BSA control or CEP. N=3 per group. (**S**) Summary of the main findings of this study. Unpaired t-test (**G, P**) or Mann-Whitney test (**I, L, N, Q, R**) was used to determine the statistical significance. All data are mean ± s.e.m. A p-value ≤ 0.05 was considered to be statistically significant.

The online version of this article includes the following figure supplement(s) for figure 6:

**Figure supplement 1.** Myeloperoxidase (MPO) expression in sebaceous gland of hair follicles from old vs young mice; effect of carboxyethylpyrrole (CEP) in vitro and in vivo on hair follicle growth.

is linked to low levels of its ligand in HFs. The stimulatory role of TLR2 signaling in HFs was demonstrated through both animal models and established human cell lines.

The lack of *Tlr2* appears to shift the balance between activating and inhibitory cues, leading to a resting phase that is approximately three times longer. This is a substantial impact considering that there are only four to five hair cycles in a mouse lifetime (*Choi et al., 2021*). The decrease in both TLR2 and its ligand observed in aging and associated conditions will inevitably impede the cyclic regeneration of HFs.

The immune system was shown to play a role in the activation of HFSCs, even in the absence of inflammation (*Ali et al., 2017*; *Castellana et al., 2014*; *Pinho and Frenette, 2019*). TLR2 expression increases at the onset of anagen when the immune response is reduced (*Paus et al., 2003*) and the HF is most susceptible to pathogens. The upregulation of TLR2 in anagen may initially have a protective function. However, high TLR2 expression in undifferentiated vs. differentiated cells underscores its role in stem cell biology. TLRs, including TLR2, have been shown to play a critical role in stem cell functions in various organs (*Lathia et al., 2008*; *Tomchuck et al., 2008*; *Trowbridge and Starczynowski, 2021*). TLRs' ligation and signaling can alter stem cell differentiation patterns (*Collins et al., 2021*; *Nagai et al., 2006*; *Trowbridge and Starczynowski, 2021*). Proinflammatory signaling can also activate HFSC proliferation, e.g., during injury (*Chen et al., 2015*; *Wang et al., 2017*). During the immune-privileged anagen phase of the hair cycle, TLR2 signaling may act as a key intrinsic factor in triggering HFSC activation.

The role of innate immunity in stem cell activation has mainly been linked to TLR3, shown to induce pluripotency in somatic cells through nuclear reprogramming (*Lee et al., 2012*; *Sayed et al., 2015*) and drive HF neogenesis after tissue damage (*Nelson et al., 2015*). In contrast, we show that TLR2 drives a rapid inflammatory response and regulates the normal hair cycle and HF regeneration/neogenesis in injury.

TLR2 promotes the hair cycle by inhibiting the BMP pathway, a key regulator of HFSC quiescence (*Hsu et al., 2011*; *Kandyba et al., 2013*; *Plikus et al., 2008*). Our study demonstrates that reducing excessive BMP signaling reactivates *Tlr2*-deficient HFSCs, revealing a novel link between TLR2, BMP

signaling, and the hair cycle. The only known instance of immune system-mediated BMP pathway inhibition occurs during the apoptosis of bulge-associated macrophages (*Castellana et al., 2014*; *Wang et al., 2017*).

The role of TLR2 in HF regeneration in both normal hair growth and wound healing emphasizes the importance of understanding the nature of TLR2 ligands mediating these responses. At the site of injury, TLR2 can be activated by pathogens or by endogenously produced ligands, such as the oxidative product of PUFA, CEP, generated in abundance during wound healing (*West et al., 2010*; *Yakubenko et al., 2018*). CEP and TLR2 are both essential for hair regeneration, and their deficiency observed in pathologies such as aging and obesity might substantially impair hair growth. Exogenous application of CEP accelerates both wound closure (*West et al., 2010*) and HF regeneration through TLR2. In addition, both TLR2 and the application of CEP diminish inhibitory BMP signaling, suggesting that CEP and other TLR2 ligands could have therapeutic value for the treatment of hair loss related to burns, traumas, and other pathologies. It is intriguing that while CEP is almost exclusively generated at sites of injury and inflammation (*Xiong et al., 2022a*; *Yakubenko et al., 2018*), HFs continuously produce it, most likely by the means of MPO, an anti-bacterial enzyme, capable of generating CEP (*Klebanoff et al., 2013*).

Contrary to the trend observed in other tissues, where the accumulation of oxidation-generated CEP increases with aging (*West et al., 2010*), HFs seem to exhibit depletion of CEP with aging. This decline in CEP levels might contribute to the reduced activity of HFSCs (*Fuchs and Blau, 2020*).

The role of CEP in TLR2-dependent HF growth and regeneration highlights the connection between oxidative stress and regenerative processes. Sustained reactive oxygen species (ROS) play a crucial role in proper regeneration, as seen in the tail amputation of *Xenopus* tadpole (*Love et al., 2013*). ROS enhance the differentiation of hematopoietic progenitors in *Drosophila* (*Owusu-Ansah and Banerjee, 2009*) and sustain self-renewal in neural stem cells (*Le Belle et al., 2011*). Additionally, ROS production in the skin has been linked to the activation of HFSCs (*Carrasco et al., 2015*). We show the underlying mechanism for these observations, where oxidation-generated CEP triggers TLR2 activation, decreases inhibitory BMP signaling, and stimulates HF growth and regeneration. TLR2 appears to serve as a common link between oxidative stress and tissue regeneration.

To summarize, our study highlights a novel role of TLR2 in promoting tissue regeneration during normal hair growth and wound healing. The identification of an endogenous TLR2 ligand produced by HFs presents a potential target for augmenting hair regeneration in the context of injury and aging, opening up new avenues for regenerative medicine.

## Materials and methods

### Mice

Inducible K15-CrePR1 mice (Stock No. 005249), TLR2-GFP reporter mice (Stock No. 031822), and TLR2$^{KO}$ mice (Stock No. 004650) were purchased from the Jackson Laboratory. *Tlr2*$^{flox/flox}$ mice with Exon3 of the *Tlr2* gene flanked by two loxP sites were described elsewhere (*McCoy et al., 2021*). HFSC-specific *Tlr2* KO (TLR2$^{HFSC-KO}$) mouse line was described previously (*Xiong et al., 2022b*). Briefly, *Tlr2*$^{flox/flox}$ mice were crossed with K15-CrePR1 mice to generate the inducible HFSC-specific *Tlr2* KO mouse line. To induce Cre recombinase activity, RU486 (Sigma) was used topically on shaved dorsal skin (1% mixed with Neutrogena Hand Cream) or via intraperitoneal injection (10 mg/ml in corn oil, 75 µg RU486 per 1 kg body weight) during the first postnatal telogen. To block BMP signaling in mouse HFs, at first postnatal telogen after applying Ru486, 200 ng of recombinant mouse Noggin (BioLegend) reconstituted in 30 µl of PBS were injected intradermally into a dorsal skin for 3–5 consecutive days. BSA in PBS was used as vehicle control.

For high-fat diet feeding studies, male WT C57BL/6J were purchased from Jackson Laboratories (Bar Harbor, ME, USA), and at 7 weeks of age, mice were either maintained on standard rodent chow or switched to a high-fat diet containing 60% of kilocalories from fat (Research Diets D12492) for an additional 15 weeks prior to tissue collection. For all animal experiments mice were randomly assigned to the groups (if it had been required), and results were evaluated in a blinded manner. All procedures were performed according to animal protocols (00002319) approved by the Cleveland Clinic IACUC committee. All surgical procedures were performed under ketamine/xylazine anesthesia followed by subcutaneous injection of a single dose of buprenorphine SR after surgery. According

to veterinarian recommendations, water with acetaminophen was provided for the next 5-7 days to minimize suffering.

## Cells

Mouse keratinocytes and HFSCs were isolated from the mouse dorsal skin as described previously (*Xiong et al., 2022b*). Briefly, isolated dorsal skin samples were trypsinized, the epidermis was scrapped, minced, and filtered through a 70 µm cell strainer to prepare primary keratinocytes single-cell suspension. To isolate HFSC, the single-cell suspension was incubated with CD34-FITC antibody (eBioscience, 11-0341-82), Alexa 647-conjugated CD49f antibody (BD Biosciences, 562494), 7-AAD (BD Biosciences, 559925), and different fluorescence minus one was used as a control. Cells were then sorted by BD FACS Aria and analyzed by Flow Jo. All the primary cells were used within 48 hr for experiments.

Human HFDPCs, mycoplasma tested, were purchased from Cell Applications, Inc (cat.# 602-05a). Human HFSCs, mycoplasma tested, were purchased from Celprogen (cat.# 36007-08). Human epidermal keratinocytes, neonatal, pooled, mycoplasma tested, were purchased from Lonza Reagents (cat.# 192906).

## Immunostaining

Mouse skin samples were harvested at indicated ages and fixed in 4% paraformaldehyde, kept in 30% sucrose for 2–3 days, followed by snap-freezing at –80°C in OCT (Fisher HealthCare, 4585). 10 µm skin sections were permeabilized, blocked, and incubated with primary antibodies followed by incubation with the corresponding secondary antibody, and mounted with an antifade mounting medium with DAPI (Vector Laboratories, H-1500-10). Images were captured on a Leica DM2500 confocal microscope and analyzed using Bitplane Imaris software (version 9.7.2) or ImageJ. Briefly, image z-stacks were loaded into Imaris to reconstruct three-dimensional images, and surface rendering was performed with default settings using the surface tool. The same background subtraction was performed on each z-stack. The green channel was used as a source channel to create surfaces for GFP$^+$ cells in the area of interest in HFs, and other channels were created based on the expression of different cell markers (e.g. CD34, Ker5) in the HFs. The overlap between GFP surface and other maker surfaces was created and visualized as the co-localized area with the co-localization module. At least 10 HFs from each mouse were used for quantification.

The following antibodies or reagents were used: Ker17 (Santa Cruz Biotechnology, sc-393002), Ker15 (ABclonal, A2660), MPO (Santa Cruz Biotechnology, sc-390109), GFP (Thermo Fisher Scientific, CAB4211), TLR2 (Santa Cruz Biotechnology, sc-21759), Ki67 (Abcam, ab16667), P-cadherin (R&D Systems, AF761-SP), CEP (Pacific Immunology), pSmad1/5/9 (Cell Signaling Technology, 13820S), β-catenin (Cell Signaling Technology, 8480), CD34 (eBioscience, 11-0341-82), CD49f (BD Pharmigen, 562473), Ker5 (BioLegend, 905903), Sox9 (Cell Signaling Technology, 82,630T), BMP7 (Proteintech, 12221-1-AP), and Nile Red (ATT BioQuest, 250730). As a negative control, we used appropriate isotype match nonimmune antibody: normal mouse IgG2b-PE (Santa Cruz Biotechnology, sc-2868), normal goat IgG control (R&D, AB-108-C), normal mouse IgG (Santa Cruz Biotechnology, sc-2025), normal rabbit IgG (Cell Signaling Technology, 2729S), normal rat IgG (Santa Cruz Biotechnology, sc-2026).

## Wound healing

Mouse wound healing procedure was performed as previously described (*West et al., 2010*; *Xiong et al., 2022b*). Briefly, an intraperitoneal injection of a ketamine/xylazine cocktail was used to anesthetize 7- to 8-week-old mice. After shaving, full-thickness wounds were made into the dorsal skin using a 6 mm biopsy punch. To examine the effect of CEP on hair regeneration after wound healing, CEP (CEP in polyethylene glycol) or vehicle (polyethylene glycol) was applied to the wounded area every day for 2 weeks. Pictures were taken at different time points to record hair regeneration around the wounded area.

## Primary keratinocyte proliferation assay

The primary keratinocytes after isolation were plated on rat tail collagen-coated plates with Epilife medium (Gibco, MEPI500CA) supplemented with EDGS (Gibco, S0125). Cells were co-cultured with

CEP or control (BSA or PBS) for 48 hr. The cell counting kit 8 (APEXbio, 269070) has been used to measure cell proliferation according to the manufacturer's protocol.

## Human HFDPC proliferation assay

HFDPCs (Cell Applications, Inc cat.# 602-05a) were cultured in HFDPC Growth Medium (Cell Applications, Inc cat.# 611-500) for 60 hr and then transferred into collagen-coated 48-well plate for 24 hr. After 24 hr cells were washed with 1× D-PBS and incubated in HFDPC Basal Medium contains no growth supplement (Cell Applications, Inc cat.# 610-500) for the next 24 hr. After starvation, cells were incubated with CEP 5 µM or with HFDPC Growth Medium (positive control), or in HFDPC Basal Medium (negative control) for 48 hr. Absorbance was read using cell counting kit 8 (ApexBio, cat.# K1018) on a microplate reader.

## Human HFSC proliferation assay

Human HFSCs (Celprogen cat.# 36007-08) were cultured in HFSC Un-differentiation Media with Serum (Celprogen cat.# M36007-08US) for 48 hr and then transferred into Undifferentiated ECM 96-Well Plates (Celprogen cat.# UD36007-08-96Well) for 24 hr. After 24 hr cells were washed with 1× D-PBS and incubated in HFSC Serum Free Un-differentiation Media (Celprogen cat.# M36007-08U) overnight. After starvation, cells were incubated for 2 hr with or without TLR2 blocking antibody (Invivogen cat.# mab2-mtlr2) followed by incubation with Pam3CSK4 (Invivogen cat.# tlrl-pms) for 24 hr. HFSC Serum Free Un-differentiation Media was used as a negative control. Absorbance was read using cell counting kit 8 (ApexBio, cat.# K1018) on a microplate reader.

## Human epidermal keratinocytes experiments

Human epidermal keratinocytes (Lonza Reagents cat.# 192906) were cultured in KGMTM Gold Keratinocyte Growth Medium (Lonza Reagents cat.# 192060) for 60 hr and then transferred into a six-well plate for 24 hr. After 24 hr media was changed, and cells were incubated with Pam3CSK4 10 µg/ml for 1 hr followed by incubation with BMP4 10 ng/ml for 1 hr.

## CEP synthesis and preparation

The structure and synthesis of CEP have been described elsewhere (*West et al., 2010*). To prepare CEP for wound healing, 250 µl CEP in PBS was mixed with 1.1 g polyethylene glycol with sonication in a 45°C water bath for 15 min followed by a strong vortex to mix well. This mixture was stored at 4°C after preparation, warmed to room temperature, and mixed again before use.

## Real-time qPCR

Total RNA from primary keratinocytes or HFSCs was isolated with RNeasy Mini Kit (QIAGEN, 74104) and reverse-transcribed into cDNA with PrimeScript RT Master Mix (Takara, RR036A). The real-time PCR was performed using iQ SYBR Green Supermix (Bio-Rad, 1708882) on the Bio-Rad cfx96 qPCR system. Target gene expression levels were normalized to internal control Rps16, and the ΔΔCt method was used to calculate fold change in gene expression. Primers can be found in *Supplementary file 1*.

## Western blot analysis

Cells were lysed with RIPA Lysis and Extraction Buffer (Thermo Scientific cat.# PI89900) buffer with protease/phosphatase inhibitor cocktail. The lysate was centrifuged at 12,000×*g* at 4°C for 15 min, boiled with Laemmli buffer for 7 min at 95°C, and transferred to PVDF membranes (Millipore). After blocking, membranes were incubated with primary antibody at 4°C overnight followed by incubation with corresponding secondary HRP-linked antibody. The following antibodies were used for western blotting: Smad1 (D59D7) XP Rabbit mAb (Cell Signaling Technology cat.# 6944), Phospho-Smad1 (Ser463/465)/Smad5 (Ser463/465)/Smad9 (Ser465/467) (Cell Signaling Technology cat.# 13,820P), NF-κB p65 (D14E12) XP Rabbit mAb (Cell Signaling Technology cat.# 8242), Phospho-NF-κB p65 (Ser536) (93H1) Rabbit mAb (Cell Signaling Technology cat.# 3033), Anti-GAPDH antibody EPR16884 Loading Control (Abcam cat.# ab181603).

## BMT and wound assay

We performed bone marrow transplant (BMT) as previously described (*West et al., 2010*) Briefly, 2-month-old male WT or TLR2[KO] mice were lethally irradiated with 9 Gy followed by tail vein injection

with $10^7$ bone marrow cells isolated from the WT donor femurs. Eight weeks after BMT, mice were subjected to wound healing assay (described above).

## RNA sequencing and data analysis

First telogen mouse dorsal skin was used for HFSC isolation by FACS. Total RNA was extracted using the RNeasy Mini Kit (QIAGEN, 74104). Sample quality assessment was performed on a Fragment Analyzer electrophoresis system (Agilent). Total RNA was normalized prior to oligo-dT capture and cDNA synthesis with SMART-Seq v4 (Takara). The resulting cDNA was quantified using a Qubit 3.0 fluorometer (Life Technologies). Libraries were generated using the Nextera XT DNA Library Prep kit (Illumina). Medium-depth sequencing (50 million reads per sample) was performed with a NextSeq 550 (Illumina) on a High Output flow cell using 75 base pairs, Paired-End run. Raw demultiplexed fastq paired-end read files were trimmed of adapters and filtered using the program skewer to throw out any with an average Phred quality score of less than 30 or a length of less than 36. Trimmed reads were then aligned using the HISAT2 aligner to the Mouse NCBI reference genome assembly version GRCm38 and sorted using SAMtools. Aligned reads were counted and assigned to gene meta-features using the program featureCounts as part of the Subread package. These count files were imported into the R programming language and were assessed for quality control, normalized, and analyzed using an in-house pipeline utilizing the limma-trend method for differential gene expression testing and the GSVA library for gene set variation analysis. The pathway analysis for differentially expressed genes with adjusted p-value<0.05 was performed using Enrichr web server https://maayanlab.cloud/Enrichr.

## Statistical analysis

Statistical analyses were performed using GraphPad Prism 9. All results are mean ± s.e.m. Shapiro-Wilk normality and lognormality test was used with n≥6. For normally distributed data, we use an unpaired two-tailed t-test to compare two groups and the one-way ANOVA followed by Dunnett's or Tukey's post hoc analysis to compare more than two groups. For non-normally distributed data and small sample size (n<6), we appraised statistical differences with the non-parametric Mann-Whitney test to compare two sample datasets and the Kruskal-Wallis test with Dunn's post hoc test for three or more groups. A p-value ≤ 0.05 was considered to be statistically significant. The sample size was calculated based on a significance level of 0.05 and power 80% (0.8).

## Acknowledgements

We thank D Nascimento and K Li for mouse colony management; T Dudiki for revision of figures; J Powers for FACS assistance; and C Nelson for proofreading. Applied Functional Genomics Core for RNA sequencing and M Kumar for data analysis. National Institutes of Health grant R01 HL145536.

## Additional information

### Competing interests

Tatiana V Byzova: Dr. Byzova has a relevant patent 9,981,018 'Compositions and Methods for Modulating Toll-Like Receptor 2 Activation'. The other authors declare that no competing interests exist.

### Funding

| Funder | Grant reference number | Author |
|---|---|---|
| National Institutes of Health | R01 HL145536 | Tatiana V Byzova |

The funders had no role in study design, data collection and interpretation, or the decision to submit the work for publication.

## Author contributions
Luyang Xiong, Irina Zhevlakova, Data curation, Investigation, Methodology, Writing – original draft; Xiaoxia Z West, Rakhilya Murtazina, Data curation, Investigation, Methodology; Detao Gao, Resources, Methodology; Anthony Horak, Resources; J Mark Brown, Writing – review and editing; Iuliia Molokotina, Investigation; Eugene A Podrez, Resources, Supervision; Tatiana V Byzova, Conceptualization, Resources, Data curation, Supervision, Funding acquisition, Investigation, Methodology, Writing – original draft, Writing – review and editing

## Author ORCIDs
Irina Zhevlakova http://orcid.org/0000-0002-8596-3329
Rakhilya Murtazina https://orcid.org/0000-0002-0819-9953
Anthony Horak https://orcid.org/0009-0009-2024-3679
J Mark Brown https://orcid.org/0000-0003-2708-7487
Tatiana V Byzova https://orcid.org/0000-0002-2615-875X

## Ethics
All procedures were performed according to animal protocols (00002319) approved by the Cleveland Clinic IACUC committee. All surgical procedures were performed under ketamine/xylazine anesthesia followed by subcutaneous injection of a single dose of buprenorphine SR after surgery. Water with acetaminophen was provided for the next 5-7 days to minimize suffering.

Reviewer #1 (Public Review): https://doi.org/10.7554/eLife.89335.3.sa1
Reviewer #3 (Public Review): https://doi.org/10.7554/eLife.89335.3.sa2
Author Response https://doi.org/10.7554/eLife.89335.3.sa3

# Additional files

## Supplementary files
• Supplementary file 1. Quantitative polymerase chain reaction (qPCR) primers.
• MDAR checklist

## Data availability
The RNAseq dataset is available in the Gene Expression Omnibus GSE179300.

The following dataset was generated:

| Author(s) | Year | Dataset title | Dataset URL | Database and Identifier |
|---|---|---|---|---|
| Xiong L, Zhevlakova I, West XZ, Gao D, Murtazina R, Horak A, Mark Brown J, Molokotina I, Podrez EA, Byzova TC | 2024 | Innate immunity controls hair regeneration and growth via BMP signaling | https://www.ncbi.nlm.nih.gov/geo/query/acc.cgi?acc=GSE179300 | NCBI Gene Expression Omnibus, GSE179300 |

The following previously published datasets were used:

| Author(s) | Year | Dataset title | Dataset URL | Database and Identifier |
|---|---|---|---|---|
| Greco V, Chen T, Rendl M, Schober M, Amalia Pasolli H, Stokes N, Cruz-Racelis JD, Fuchs E | 2009 | Expression data from sorted follicle populations in the 2nd telogen to anagen transition | https://www.ncbi.nlm.nih.gov/geo/query/acc.cgi?acc=GSE15185 | NCBI Gene Expression Omnibus, GSE15185 |

*Continued on next page*

*Continued*

| Author(s) | Year | Dataset title | Dataset URL | Database and Identifier |
|---|---|---|---|---|
| Morinaga H, Mohri Y, Grachtchouk MA, Asakawa K, Matsumura H, Oshima M, Takayama N, Kato T, Nishimori Y, Sorimachi Y, Takubo K, Suganami T, Iwama A, Iwakura Y, Dlugosz AA, Nishimura EK, Andrzej AD, Nishimura EK | 2021 | Obesity accelerates hair thinning in stem cell-centric converging mechanism | https://www.ncbi.nlm.nih.gov/geo/query/acc.cgi?acc=GSE131958 | NCBI Gene Expression Omnibus, GSE131958 |

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

# Appendix 1

## Appendix 1—key resources table

| Reagent type (species) or resource | Designation | Source or reference | Identifiers | Additional information |
|---|---|---|---|---|
| Strain, strain background (*Mus musculus*) | K15-CrePR1 | The Jackson Laboratory | Cat.# 005249; RRID:IMSRJAX:005249 | RU 486-inducible Cre recombinase driven by the mouse keratin complex 1, acidic, gene 15 promoter. When induced, Cre activity is observed in epithelial stem cells in the bulge region of the hair follicle |
| Strain, strain background (*Mus musculus*) | TLR2-GFP | The Jackson Laboratory | Cat.# 031822; RRID:IMSR_JAX:031822 | *Tlr2KI* knock-in mice have an HA tag and an *IRES-EGFP* sequence placed at the 3' end of the Toll-like receptor 2 (*Tlr2*) gene |
| Strain, strain background (*Mus musculus*) | TLR2KO | The Jackson Laboratory | Cat.# 004650; RRID:IMSR_JAX:004650 | Global *Tlr2* KO |
| Strain, strain background (*Mus musculus*) | *Tlr2*$^{flox/flox}$ | Taconic Laboratory | c57BL/6NTacTlr2^tm3243Arte | loxP sites on either side of exon 3 of the targeted TLR2 gene |
| Strain, strain background (*Mus musculus*) | TLR2$^{HFSC-KO}$ | Described previously; *Xiong et al., 2022b* | c57BL/6NTacTlr2^tm3243Arte -B6;SJL-Tg(Krt1-15-cre/PGR)22Cot/J | RU 486-inducible hair follicle stem cells-specific *Tlr2* KO |
| Strain, strain background (*Mus musculus*) | C57BL/6J | The Jackson Laboratory | Cat.# 000664 RRID:IMSR_JAX:000664 | |
| Cell line (*Mus musculus*) | Skin keratinocytes | Described previously; *Xiong et al., 2022b* | | Freshly isolated from the mouse dorsal skin of WT and TLR2KO mice |
| Cell line (*Mus musculus*) | Hair follicle stem cells | Described previously; *Xiong et al., 2022b* | | Freshly isolated from the mouse dorsal skin of WT and TLR2$^{HFSC-KO}$ mice |
| Cell line (human) | Hair follicle dermal papilla cells | Cell Applications, Inc | Cat.# 602-05a | Normal human scalp hair follicle papilla cells |
| Cell line (human) | Hair follicle stem cells | Celprogen | Cat.# 36007-08 | Human frontal region scalp extracted from hair follicle bulge |
| Cell line (human) | Epidermal keratinocytes, neonatal, pooled | Lonza Reagents | Cat.# 192906 | Cryopreserved normal human epidermal keratinocytes from pooled donors |
| Antibody | Mouse monoclonal anti-keratin 17 | Santa Cruz Biotechnology | Cat.# sc-393002- AF647; RRID:AB_2893006 | IF 1:200 |
| Antibody | Rabbit polyclonal anti-keratin 15 | Abclonal | Cat.# A2660; RRID:AB_2764526 | IF 1:100 |
| Antibody | Mouse monoclonal anti-myeloperoxidase | Santa Cruz Biotechnology | Cat.# sc-390109; RRID:AB_2892996 | IF 1:100 |
| Antibody | Mouse monoclonal anti-TLR2 | Santa Cruz Biotechnology | Cat.# sc-21759 RRID:AB_628363 | IF 1:100 |
| Antibody | Rabbit polyclonal anti-GFP | Thermo Fisher Scientific | Cat.# sc-390109; RRID:AB_10709851 | IF 1:100 |
| Antibody | Rabbit monoclonal anti-Ki67 | Abcam | Cat.# ab16667; RRID:AB_302459 | IF 1:250 |
| Antibody | Goat polyclonal anti-P-cadherin | R&D Systems | Cat.# AF761-SP; RRID:AB_355581 | IF 1:50 |
| Antibody | Rabbit polyclonal anti-CEP | Pacific Immunology | Custom | IF 1:200 |

*Appendix 1 Continued on next page*

*Appendix 1 Continued*

| Reagent type (species) or resource | Designation | Source or reference | Identifiers | Additional information |
|---|---|---|---|---|
| Antibody | Rabbit polyclonal anti-β-catenin | Cell Signaling Technology | Cat.# 8480; RRID:AB_11127855 | IF 1:80 |
| Antibody | Rat monoclonal anti-CD34 | eBioscience | Cat.# 11-0341-82; RRID:AB_465021 | IF 1:200 FACS 1 µg/test |
| Antibody | Rat monoclonal anti-CD49f | BD Biosciences | Cat.# 562473; RRID:AB_11153684 | IF 1:100 FACS 5 µl/test |
| Antibody | Rabbit monoclonal anti-Sox9 | Cell Signaling Technology | Cat.# 82,630T; RRID:AB_2665492 | IF 1:200 |
| Antibody | Chicken polyclonal anti-keratin 5 | BioLegend | Cat.# 905903; RRID:AB_2721742 | IF 1:200 |
| Antibody | Rabbit polyclonal anti-BMP7 | Proteintech | Cat.# 12221-1-AP; RRID:AB_2063960 | IF 1:200 |
| Antibody | Rabbit monoclonal anti-pSmad1/5/9 | Cell Signaling Technology | Cat.# 13,820P; RRID:AB_2493181 | IF 1:200 WB 1:1000 |
| Antibody | Anti-murine TLR2 (clone T2.5) Detection and Neutralizing mouse monoclonal | Invivogen | Cat.# mab2-mtlr2 RRID N/A | Blocking experiment 0.66 µg/ml |
| Antibody | Smad1 (D59D7) XP Rabbit monoclonal | Cell Signaling Technology | Cat.# 6944 | WB 1:1000 |
| Antibody | NF-κB p65 (D14E12) XP Rabbit monoclonal | Cell Signaling Technology | Cat.# 8242 | WB 1:1000 |
| Antibody | Phospho-NF-κB p65 (Ser536) (93H1) Rabbit monoclonal | Cell Signaling Technology | Cat.# 3033 | WB 1:1000 |
| Antibody | Anti-GAPDH antibody EPR16884 Loading Control Rabbit monoclonal | Abcam | Cat.# 181603 | WB 1:6000 |
| Antibody | Anti-rabbit IgG, HRP-linked Antibody goat anti-rabbit IgG Polyclonal | Cell Signaling Technology | Cat.# 7074S | WB 1:3000 |
| Antibody | Normal mouse IgG2b-PE isotype control | Santa Cruz Biotechnology | Cat.# sc-2868 RRID:AB_737259 | According to immune antibody concentration |
| Antibody | Normal goat IgG isotype control | R&D | Cat.# AB-108-C RRID:AB_354267 | According to immune antibody concentration |
| Antibody | Normal mouse IgG isotype control | Santa Cruz Biotechnology | Cat.# sc-2025 RRID:AB_737182 | According to immune antibody concentration |
| Antibody | Normal rabbit IgG isotype control | Cell Signaling Technology | Cat.# 2729S RRID:AB_1031062 | According to immune antibody concentration |
| Antibody | Normal rat IgG isotype control | Santa Cruz Biotechnology | Cat.# sc-2026 RRID:AB_737202 | According to immune antibody concentration |
| Antibody | Chicken IgY Isotype Control | Novus Biologicals | Cat.# AB-101-C RRID:AB_354263 | According to immune antibody concentration |
| Antibody | Goat anti-Rat IgG Polyclonal (H+L) Cross-Adsorbed Secondary Antibody, Alexa Fluor 594 | Invitrogen | Cat.# A-11007 RRID:AB_10561522 | IF 1:300 |

*Appendix 1 Continued on next page*

*Appendix 1 Continued*

| Reagent type (species) or resource | Designation | Source or reference | Identifiers | Additional information |
|---|---|---|---|---|
| Antibody | Goat anti-Rabbit IgG Polyclonal (H+L) Cross-Adsorbed Secondary Antibody, Alexa Fluor 488 | Thermo Fisher Scientific | Cat.# A-11008 RRID:AB_143165 | IF 1:300 |
| Antibody | Goat anti-Rat IgG Polyclonal (H+L) Cross-Adsorbed Secondary Antibody, Alexa Fluor 488 | Invitrogen | Cat.# A-11006 RRID:AB_2534074 | IF 1:300 |
| Antibody | Alexa Fluor Plus 594 Goat anti-rabbit Polyclonal Secondary Antibody | Thermo Fisher Scientific | Cat.# A-32740 RRID:AB_2762824 | IF 1:300 |
| Antibody | Goat anti-Mouse IgG (H+L) Cross-Adsorbed Polyclonal Secondary Antibody, Alexa Fluor 568 | Thermo Fisher Scientific | Cat. # A-11004 RRID:AB_2534072 | IF 1:300 |
| Antibody | Goat anti-Mouse IgG (H+L) Cross-Adsorbed Polyclonal Secondary Antibody, Alexa Fluor 488 | Thermo Fisher Scientific | Cat.# A-11001 RRID:AB_2534069 | IF 1:300 |
| Antibody | Donkey anti-Goat IgG (H+L) Cross-Adsorbed Polyclonal Secondary Antibody, Alexa Fluor 594 | Thermo Fisher Scientific | Cat.# A-11058 RRID:AB_142540 | IF 1:300 |
| Antibody | Goat anti-Chicken IgY (H+L) Secondary Antibody, Polyclonal Alexa Fluor 488 | Invitrogen | Cat.# A-11039 RRID:AB_2534096 | IF 1:300 |
| Other | DAPI Solution | BD Biosciences | Cat#564907 RRID:AB_2869624 | Fluorescent stain IF 1:300 |
| Other | Nile Red | ATT BioQuest | Cat.# 22190 | Lipophilic stain IF 10 µM |
| Other | 7-AAD | BD Biosciences | Cat.# 559925 RRID:AB_2869266 | Membrane impermeant dye 0.25 µg/test |
| Sequence-based reagent | TLR2_F | This paper | PCR primers | TCTAAAGTCGATCCGCGACAT |
| Sequence-based reagent | TLR2_R | This paper | PCR primers | CTACGGGCAGTGGTGAAAACT |
| Sequence-based reagent | BMP7_F | This paper | PCR primers | ACGGACAGGGCTTCTCCTAC |
| Sequence-based reagent | BMP7_R | This paper | PCR primers | ATGGTGGTATCGAGGGTGGAA |
| Sequence-based reagent | BMP2_F | This paper | PCR primers | GGGACCCGCTGTCTTCTAGT |
| Sequence-based reagent | BMP2_R | This paper | PCR primers | TCAACTCAAATTCGCTGAGGAC |
| Sequence-based reagent | BMPr1A_F | This paper | PCR primers | AACAGCGATGAATGTCTTCGAG |
| Sequence-based reagent | BMPr1A_R | This paper | PCR primers | GTCTGGAGGCTGGATTATGGG |
| Sequence-based reagent | NFkB2_F | This paper | PCR primers | GGCCGGAAGACCTATCCTACT |

*Appendix 1 Continued on next page*

*Appendix 1 Continued*

| Reagent type (species) or resource | Designation | Source or reference | Identifiers | Additional information |
|---|---|---|---|---|
| Sequence-based reagent | NFkB2_R | This paper | PCR primers | CTACAGACACAGCGCACACT |
| Sequence-based reagent | IL1b_F | This paper | PCR primers | GCAACTGTTCCTGAACTCAACT |
| Sequence-based reagent | IL1b_R | This paper | PCR primers | ATCTTTTGGGGTCCGTCAACT |
| Sequence-based reagent | IL6_F | This paper | PCR primers | TAGTCCTTCCTACCCCAATTTCC |
| Sequence-based reagent | IL6_R | This paper | PCR primers | TTGGTCCTTAGCCACTCCTTC |
| Chemical compound, drug | Pam3CSK4 | Invivogen | Cat.# tlrl-pms | 10 µg/ml |
| Chemical compound, drug | Recombinant Human BMP-4 Animal-Free Protein | R&D Systems | Cat.# AFL314E-010 | 20 ng/ml |
| Chemical compound, drug | CEP (carboxyethylpyrrole) | Custom | Custom | Cell experiments 2.5–5 µM Skin treatment 5 µg/ml |
| Software, algorithm | Imaris V9.7.2 | Bitplane | | |
| Software, algorithm | ImageJ, Fiji V1.53t | National Institutes of Health | | |
| Software, algorithm | GraphPad Prism 9 | GraphPad by Dotmatics | | |
| Software, algorithm | Flow Jo | Becton, Dickinson & Company | | |

