## [Editor Report · eLife assessment]

Toll like receptor 2 (TLR2) signaling has traditionally been viewed a surface protein that induces innate immune responses and improves acquired immunity. Here, the authors suggest a different role for TLR2 in the hair cycle. By using a Cre reporter that is largely, but not solely active in hair follicle stem cells, the authors conditionally delete Tlr2 in mice and report that BMP signaling is sustained and hair cycle entry is delayed. Delving further, the authors identify CEP (2-ω-carboxyethyl pyrrole) as an endogenous ligand of TLR2 in hair follicle stem cell regulation. Although a role for TLR2 signaling in hair follicle stem cells is potentially novel and **important**, the reviewers remain in consensus that evidence presented in two significant areas continues to be **incomplete**: (1) where TLR2 and CEP are expressed and how specific is their expression to the hair follicle stem cells; (2) whether as the authors suggest, TLR2 functions by regulating BMP signaling in the stem cell niche of the hair follicle.

---

## [Referee Report · Reviewer #1 (Public Review)]

Summary:

In this manuscript by Xiong L et al., the authors have uncovered an important link between innate immune signaling and hair regeneration. The authors provide convincing evidence supporting the critical roles of TLR2 in sensing CEP levels in hair follicles, counteracting the action of BMP signaling, and facilitating the activation of HFSCs during the hair cycle and wound repair. Importantly, the authors also propose that decreased CEP production and TLR2 expression might be factors contributing to the decreased hair regeneration associated with aging.

Strengths:

The experiments in this manuscript are well-designed and presented. The authors provided extensive evidence supporting the roles of TLR2 signaling in regulating hair follicle stem cell functions. Importantly, the findings from this paper could have sustained impacts on our understanding of the roles of innate immunity in regulating tissue regeneration in the absence of inflammation.

Weaknesses:

1. The central conclusion of this study is that the activation of TLR2 can suppress BMP signaling. However, the molecular link between TLR2 and BMP signaling is still missing. Given the importance of this finding, it would be intriguing to further investigate how TLR2 activation suppresses BMP signaling. A better characterization of the molecular-level interaction between TLR2 and BMP signaling can further enhance the impact of this study.

2. The authors imply that the decreased CEP level in aged mice could lead to deficient TLR2 signaling, which could further cause aging-associated hair regeneration defects. But this has not been demonstrated. What are the BMPs and pSmad1/5 levels in aged skin? Another important experiment to confirm the importance of this link during aging would be to inject CEP into the aged skin and examine whether this could restore hair regeneration in aged mice.

3. The impacts of CEP/TLR2 on proliferation of keratinocytes is still weak. How much of this effect is a result of NFkB activation, and how much is simply due to inhibiting BMP signaling?

Updated comments on the revised manuscript:

The authors have addressed my previous questions.

---

## [Referee Report · Reviewer #3 (Public Review)]

Summary:

In the manuscript by Xiong and colleagues, the roles of TLR2 in hair follicle cycle regulation were investigated. By analyzing published dataset and using immunostaining and transgenic TLR2-GFP reporter mice, the authors showed that TLR2 expression is increased in the late telogen compared to the early telogen, implying that it is important for the transition between telogen to anagen hair cycle. They found that the genetic deletion of Tlr2 in hair follicle stem cells delays hair cycle entry in both homeostatic and wound-induced hair follicle regeneration. In addition, they found that CEP is an endogenous TLR2 activating ligand and triggers the progression of hair cycle in a TLR2-dependent manner. Mechanistically, the activation of TLR2 signaling antagonizes BMP signaling which is critical for the maintenance of hair follicle stem cell quiescence. Clinically, they showed that TLR2 expression is decreased in aging and high-fat diet condition, suggesting that the dysfunctional regulation of TLR2 pathway is responsible for age-related and obesity-related hair thinning and hair loss phenotypes.

Strengths:

Overall, this study presents the role and mechanism of TLR2 in regulating hair follicle regeneration. The functional interrogation parts using HFSC-specific TLR2 genetic deletion is solid, and an endogenous regulator, CEP, is identified.

Weaknesses:

1.

- In SFig1A, the IF staining of TLR2 and Tlr2-GFP expression seem almost 100% co-localized, which is not usual experimentally.

- In Fig 2J, the relative expression levels of Tlr2 in anagen, telogen, catagen HFSCs were tested. But it is just relative comparison and does not mean whether the expression level is meaningful or not. To make this convincing, adding other cell types such as dermal fibroblasts and immunes to the comparison as negative and positive controls would be a good idea.

- In Fig 2K, the expression of Tlr2 is comparable or a bit lesser in epidermal cells and HFSCs, but the expressions of TLR2 (IF) and Tlr2-GFP in epidermal cells have not been presented at all in the manuscript. As the authors used K15-CrePR1 mice to delete Tlr2 in HFSCs specifically, showing TLR2 IF staining in TLR2-HFSC-KO mice would be nice evidence of significant expression of TLR2 in HFSCs. (still TLR2 expression in epidermis, but no TLR2 expression in HFSCs).

- In Fig 1B, it is still unclear whether TLR2 staining is in epithelial cell or in dermal cells. TLR2 staining patterns in Fig 1B, SFig 1A, and rebuttal seem different. In Fig S1B and rebuttal, TLR2 expression in HFSCs, HG, DP cells, but in Fig 1B, most of HG and DP cells are not TLR2+.

- Together, this reviewer still does not think that there is a clear and solid evidence of Tlr2 expression in HFSCs. Searching the Tlr2 expression in published bulk and single cell RNA-seq dataset would be helpful.

1.

- In SFig 4B, C, the activation of BMP signaling was hindered by TLR2 signaling activation by PAM3CSK4. But it is in vitro data, and cultured HFSCs are different from in vivo HFSCs, and particularly the changes of HFSCs from quiescence to activation can hardly be recapitulated in vitro.

- In Fig 4H, it is curious that in TLR2-HFSC-KO mice, P21 HFSCs showed no pSMAD1/5/9, but it is increased in P24.

- Also, it is wondered that if ID1 and ID2, key target genes, are increased in TLR2-HFCS-KO.

- The author suggested that BMP7 is a key connection between TLR2 signaling and BMP signaling. It is curious whether BMP7 is a direct target of TLR2 pathway? Are there Nfkb (putative) binding sites in cis-regulatory regions of BMP7?

1.

- In Fig 6C, CEP expression is close to hair follicle in both anagen and telogen. Also, in Telogen, CEP expression is strong and very close to HFSCs. But In rebuttal Fig 2, CEP is localized to sebaceous gland, where MPO, a CEP producing enzyme, is expressed. Which one is correct? Also, if CEP is strongly expressed in Telogen (Fig 6C), how can HFSCs stay in quiescence with decreased BMP signaling?

---

## [Author Response]

The following is the authors’ response to the original reviews.

To the reviewers.We appreciate a detailed and deep review of our manuscript. Below are our comments and responses. Many requested data are present in the Supplementary figures of the manuscript. There seem to be two main concerns: one regarding the evidence of TLT2 expression in HFSCs; and second, regarding CEP/TLR2. As detailed below, we utilized 3 different methods to document TLR2 expression: TLR2-reporter mouse, staining for TLR2 and qPCR of isolated cells for TLR2. The source (the data are in Supplementary Fig. 5A, B and in references below) and nature of CEP (it is not a protein, but metabolic product of Polyunsaturated acid DHA oxidation by MPO amongst other ROS sources) are also explained below.1. “The expression analysis of TLR2 is questionable. Many of the conclusions about the level of target genes are based on quantifying fluorescence intensity in microscopy images (e.g., TLR2 level in young or aged mice, BMP7 levels in mice with/without TLR2 KO). This could be strengthened by using qPCR to measure gene expression levels in FACS-sorted HFSCs, which would provide more accurate quantification. Additionally, the authors should test if the TLR2 antibody used is valid.”

In most instances we have used TLR2 reporter mouse, which presents an advantage over immunostaining. Fig.2 (A-H) shows expression of TLR2 reporter, not the staining with TLR2 abs. For selected experiments we utilized immunostaining with anti- TLR2 (Santa Cruz Biotechnology, sc-21759) antibody, which has been validated in our previous publication (see Michael G. McCoy and all. Endothelial TLR2 promotes proangiogenic immune cell recruitment and tumor angiogenesis. // Sci Signal. 2021 Jan 19; 14(666): eabc5371/doi: 10.1126/ scisignal.abc5371). In Fig.S2E of that manuscript we validated these abs using a knockout of TLR2. In the current paper, we further validate anti-TLR2 abs by showing its co-localization with the TLR2-GFP reporter (Fig. S1A).

We then confirmed reporter and immunostaining data by qPCR showing Tlr2 expression in FACS-purified mouse HFSCs in anagen, telogen, and catagen (Fig.2J), in mouse epidermal cells and FACS-purified HFSCs (Fig.2K), and FACS-purified HFSCs isolated from Control and TLR2HFSC-KO mice (Fig.4E).

As for the mechanistic link between TLR2 and BMP signaling was identified using RNAseq on FACS-purified HFSCs (supplementary Fig.4), then verified using qPCR (Fig.4E shows Bmp7,Bmp2, Bmpr1a ) and only then immunohistochemistry staining for BMP7 and phosphoSMAD1/5/9 was used (Fig.4A-D, F-H). Note that the large body of requested evidence is presented in Supplementary data. Other mechanistic links shown using qPCR include Nfkb2, Il1b, Il6, and Bmp7 in FACS-purified mouse HFSCs treated with BSA control or CEP (Fig.6Q,6R).

“As the reviewers note, it is not clear whether the TLR2+ signal is located at the basal side of bulge stem cells, basement membrane underlying bulge stem cells, or dermal sheath cells encapsulating bulge structure. Co-staining with basement membrane markers such as collagen and laminin or HFSC basal side membrane markers such as Itga6, Itgb1, and Itgb4 will clarify this. In addition, showing the expression pattern of TLR2 in full skin including epidermis and dermis would be helpful. As TLR2 is highly expressed in immune cells or blood endothelial cells, if the antibody staining is valid, strong positive signals should present in the cells. Moreover, testing the TLR2 antibody in Tlr2 knock-out mouse tissues would be an appropriate control experiment.”

Once again, in most instances we have used not the staining for TLR2 but TLP2 reporter mouse (Fig.2 legend). Anti-TLR2 abs have been verified in TLR2 KO as described above. Fig.2K shows comparison of Tlr2 mRNA expression in mouse epidermal cells to FACS-purified HFSCs by qPCR.

TLR2 signal is detected in several cell types within the hair follicle as well as in dermal cells surrounding the hair follicles, such as lymphocytes, resident tissue macrophages, fibroblast, and fibroblast precursors, etc. (https://www.proteinatlas.org/ENSG00000137462-TLR2/single+cell+type). In Author response image 1 below, white arrows point to the TLR2-positive cells around the hair follicle. In our paper, we focus on HFSC TLR2 and use the respective inducible tissue specific TLR2 KO. The contribution of TLR2 on other cell types can be assessed by the comparison of the phenotypes of global TLR2 KO, TLR2 KO-WT bone marrow chimeras and HFSC-specific TLR2 KO. The results are presented in both, main and supplementary figures (Fig.5D-I and SFig.5I-K) shows global TLR2 KO, Fig.6H-I, SFig.5G-h shows bone marrow chimeras and Figs.3,4, 5 (J-M), Fig.5 (J-N) shows the main focus, HFSC-TLR2 KO. Overall, the phenotype (delay of hair regeneration after wounding) seems to be the strongest in TLR2 KO, whereas bone marrow chimeras and HFSCs phenotypes are comparable. Thus, TLR2 on bone marrow derived cells complements the main role for TLR2 on HFSCs.

**Author response image 1. sa3fig1:** Staining for TRLR2 (white), DAPI (blue) and Keratin 17 (purple) is shown.

“The increase in expression of TLR2 during the hair follicle stem cell activation should be documented by FACS and/or qPCR. This is important because as noted by one of the reviewers.”

While original observation was done using both, a TLR2 reporter mouse and immunostaining, the data were confirmed by qPCR showing Tlr2 mRNA expression in FACS-purified mouse HFSCs in anagen, telogen, and catagen (Fig.2J).

“In Fig 1D, the authors mentioned that they re-analyzed published RNA-seq data (Greco et al., 2009) to show the increase of Tlr2 and Tlr6 expression in late telogen compared to early telogen. However, there is no RNA-seq data in that paper, but only microarray data of bulge vs HG comparison and dermal papillae cells (DP) in early, mid, late Telo. If the authors used DP data to show the increase of Tlr2 transcripts in late Telo, the analysis is completely wrong and has to be corrected. The problem is compounded by the fact that in other published HFSC RNA-seq datasets (Yang et al., Cell, 2017, Adam et al., Nature Cell Biology, 2020), the expression levels of Tlr2 and Tlr6 are very low (below 5 TPM). In Fig 1G, the authors also re-analyzed Morinaga et al., 2021 data to show the reduction of Tlr2 expression in HFSCs in high-fat diet mice. However, in the raw data of Morinaga et al., 2021 (GSE169173), Tlr2 expression FPKM values are below 1 in both normal diet and high-fat diet samples, which are too low to perform comparative analysis and are not statistically meaningful. Like Tlr2, the expressions of Tlr1 and Tlr6, which form heterodimer with TLR2, are almost 0. Thus, the authors should revisit the dataset and revise their analysis and conclusion.”To document the existence of Tlr2 and Tlr6 expression in HFSCs, the authors should perform RNR-seq-based gene expression analysis by themselves. Otherwise, the authors' TLR2 expression analyses in Fig 1 are not convincing. These are serious issues that the authors will want to rectify so that eLIFE readers will not discount their findings and importance.”

It is correct, we analyzed a published array, not RNAseq data (Greco et al., 2009) using GEO2R tool which allowed us to compare the mRNA expression levels between early, middle, and late telogen in bulge CD34 positive cells. We changed the “RNA-seq” (the term was used incorrectly) to “RNA microarray” in the main text.

In our manuscript, TLR2 expression is documented not only in Fig.1, but also in Fig.2 and S.Fig.1. We utilized 3 different methods to document TLR2 expression: TLR2-reporter mouse, staining for TLR2 and qPCR of isolated cells for TLR2. Fig.2K shows comparison of Tlr2 mRNA expression in mouse epidermal cells to FACS-purified HFSCs by qPCR to document increased TLR2 expression on HFSCs. Likewise, Fig.2J shows qPCR for TLR2 on HFSC during various phases of hair growth.

“In Fig 2, to support the expression of Tlr2 in HFSCs, the authors utilized TLR2-GFP mice and showed the strong GFP expression in HFSCs, hair bulb, and ORS. However, as the expression data in Fig 1 are questionable, the GFP reporter data should be carefully analyzed with proper control experiments. For example, although TLRs are highly expressed in immune cells and endothelial cells, which are abundantly present in skin, Fig 2 data did show the GFP expression in these cells. Instead, the GFP signals looked very specific to epithelial compartments, which is odd. Again, to convince readers, the authors should provide more comprehensive analyses of expression patterns of TLR2-GFP mice in skin. Also, if the TLR2-GFP signals faithfully reflect the actual expression of Tlr2 mRNA, the GFP signals should increase in late telogen compared to early telogen. The authors should check whether TLR2-GFP expression follows this pattern.”

The specificity of TLR reporter was characterized in Price et al. , 2018. A Map of Toll-like Receptor Expression in the Intestinal Epithelium Reveals Distinct Spatial, Cell Type-Specific, and Temporal Patterns. Immunity, 49. Thus, TLR2 reporter mouse is well characterized (https://www.ncbi.nlm.nih.gov/pmc/articles/PMC6152941/) and represents one of the best available tools to show TLR2 expression.

Expression of TLR2 on endothelial cells and validation of anti-TLR2 abs was performed in McCoy et al, Science Signaling as mentioned above. Also as discussed above we show a strong correlation between TLR2-GFP reporter expression and TLR2 expression using coimmunostaining with GFP and TLR2 antibodies with appropriate isotype-match non-immune antibodies as negative controls.

There is no doubt that TLR2 is expressed on immune, endothelial and epithelial cells. According to the Human Protein Atlas, TLR2 expression is identified in skin fibroblasts, keratinocytes, melanocytes, etc., so our findings are well supported by the literature (https://www.proteinatlas.org/ENSG00000137462-TLR2/single+cell+type). Indeed, we detected TLR2 in cells surrounding the hair follicle (see the pictures above). TLR2 signal was detected in nearly all niches of hair follicles including the CD34-positive cells.

In Fig.S1 we demonstrated an increased level of TLR2 in the late (competent) telogen compared to the early (refractory) telogen using immunostaining for TLR2-GFP. The results mirrored published RNA-array data in Fig.1D. Again, reporter and immunostaining results have been validated by qPCR for TLR2.

The levels of TLR2 might be heavily influences by the environment, i.e. pathogens availability. In this regard, note that mice for this study were kept in normal, not pathogen-free conditions.

“Overall, the existence of Tlr2 expression in HFSCs is still questionable. Without resolving these, genetic deletion of Tlr2 in HFSCs cannot be rationalized.”

In our manuscript, TLR2 expression is documented not only in Fig.1, but also in Fig.2 and S.Fig.1. We utilized 3 different methods to document TLR2 expression: TLR2-reporter mouse, staining for TLR2 and qPCR of isolated cells for TLR2. Besides these data, we show the functional responses to canonical TLR2 ligand, PAM3CSK4, and previously characterized endogenous ligand, CEP, using proliferation, western blotting and many other approaches. In numerous immunostainings we show co-localization of TLR2 and CD34 (Fig.2) using IMARIS surface rendering and colocalization tools. Our conclusions are further supported by published results as discussed above.

1. “The central conclusion of this study is that the activation of TLR2 can suppress BMP signaling; however, the molecular link between TLR2 and BMP signaling is still missing. Given the importance of this finding, it would be intriguing to further investigate how TLR2 activation suppresses BMP signaling. A better characterization of the molecular-level interaction between TLR2 and BMP signaling can further enhance the impact of this study.-The published dataset should be re-analyzed, as some images and their quantification do not appear to be matched. Representative images should be used.”“In Fig 4, the authors propose that the activation of TLR2 pathway inhibits the BMP signaling pathway, which makes HFSCs quiescent. In TLR2-HFSC-KO, the authors showed that BMP7 is increased and pSMAD1/5/9 is sustained. The increase in BMP7 expression and SMAD activation should be demonstrated by additional assays. Are SMAD target genes activated in the cKO mice?”

This mechanistic link between TLR2 and BMP was originally identified by RNAseq, confirmed by qPCR and then by immunostaining for both, BMP7 and BMP pathway activation based on phosphoSMAD1/5/9 levels. The connection to BMP pathway was also shown by western blotting (S.Fig.4B,C). The rescue experiments have been performed using Noggin injections. According to our data, numerous SMAD target genes are upregulated in TLR2-HFSC-KO, such as Kank2, Ptk2b, Scarf2, Camk1, Dpysl2, as well as BMP2 and BMP7, and these changes were confirmed by qPCR analysis in Fig.4E. Additional evidence is shown in Fig.6, which demonstrates that endogenous TLR2 ligand, CEP-carboxyethylpyrrole, acts by a similar, BMP-dependent pathway. Also, Supplemental Fig.4 adds more details to this link. SFig.4B,C shows that TLR2 activation by canonical ligand PAM3CSK4 inhibits pSMAD levels induced by BMP (western blot is shown). At the same time, as anticipated PAM3CSK4 upregulated NFkB, however, little of no effect of BMP stimulation on NFkB is observed. To summarize: TLR2 affects both, BMP7 production and BMP induced downstream signaling judged by PhosphoSMADs. The later connection appears to go in one direction: TLR2 signaling affects BMP-induced pSMADs, however, BMP signaling does not seem to substantially change TLR2-dependent NFkB. We plan to delve into the intersection of these important pathways in future.

“Functionally, downregulation of BMP signaling by injecting Noggin, a BMP antagonist, in TLR2HFSC-KO mice induces HFSC proliferation. These functional data are solid. However, it is still curious how TLR2 signaling interact with BMP pathway molecularly. Is it transcriptional regulation or translational regulation? Perhaps, RNA-seq analysis of TLR2HFSC-KO could give some hints to answer this question. Furthermore, checking out other signaling pathways such as WNT/LEF1 and pCREB, which are important for hair cycle activation and NFkB, a downstream effector of TLR signaling would be helpful to interrogate mechanistic insights.”

As discussed above, TLR2 affects both, BMP7 production and BMP-induced downstream signaling judged by PhosphoSMADs. The later connection appears to go in one direction: TLR2 signaling affects BMP-induced pSMADs, however, BMP signaling does not seem to substantially change TLR2-dependent NFkB.

Indeed, in addition to BMP signaling, the Wnt signaling and β-catenin stabilization within HFSCs, known to trigger their activation (Deschene et al., 2014). However, this axis remained unchanged upon TLR2HFSC-KO (as shown in Supplementary Fig. 4J). There were several published reports on the crosstalk between TLR and BMP signaling such as (doi: 10.1089/scd.2013.0345. Epub 2013 Nov 7) showing that activation of TLR4 inhibits BMP-induced pSMAD1/5/8 and this connection requires NFkB. We probed NfkB activation, please, see the responses above.

However, we were not able to detect substantial effect of NFkB inhibition on BMP signaling in hair follicles (not shown).

1. “The function of CEP, a proposed endogenous ligand of TLR2, is still not clear. The authors imply that the decreased CEP level in aged mice could lead to deficient TLR2 signaling, which could further cause aging-associated hair regeneration defects. But this has not been demonstrated. What are the BMPs and pSmad1/5 levels in aged skin? Another important experiment to confirm the importance of this link during aging would be to inject CEP into the aged skin and examine whether this could restore hair regeneration in aged mice. Does CEP activate hair cycling during the endogenous pathway? What might be the source of CEP? Does CEP treatment activate BMP7 signaling? The authors should clarify these issues. The authors suggested that CEP is an endogenous ligand of TLR2, and administration of CEP induces hair cycle entry in a TLR2dependent manner. How potent is CEP in terms of HFSC activation? In Fig 6Q, CEP increases the expression of Nfkb2, Il1b, and Il6, but the fold changes are marginal. Also, if CEP is a critical ligand, the loss of CEP by a genetic deletion or a pharmacological inhibition should result in the delay of hair cycle entry. Furthermore, the source of CEP expression is curious. Is it expressed by HFSCs or dermal fibroblast or immune cells? Finally, comparing the effect of CEP to the effect of other bacterial origin Tlr2 ligands such as heat killed bacteria, purified microbial cell-wall components, and synthetic agonists (Pam3CSK4) would be helpful. It is curious if HFSC directly senses the bacterial materials and triggers hair follicle regeneration or are indirectly directed by immune cells and endothelial cells, which could be primary sensor.”

CEP is not a protein, it is an oxidative stress-generated metabolite of polyunsaturated fatty acid, DHA (https://www.ncbi.nlm.nih.gov/pmc/articles/PMC5360178/), thus, it is impossible to generate a knockout of this molecule. As demonstrated in previous publications(https://www.ncbi.nlm.nih.gov/pmc/articles/PMC2990914/,https://pubmed.ncbi.nlm.nih.gov/34871763/) CEP serves as a critical endogenous ligand supporting TLR2 signaling in the absence of pathogens. While other TLR2 endogenous ligands, such as HMGBs or HSPs exist (https://www.ncbi.nlm.nih.gov/pmc/articles/PMC4373479/), CEP binds to TLR2 directly, and its generation is aided by MPO (myeloperoxidase) amongst other peroxidases and sources of reactive oxygen/nitrogen species. MPO (produced by immune cells amongst others) serves as an innate immunity response against pathogens, but it also generates CEP adducts (https://www.ncbi.nlm.nih.gov/pmc/articles/PMC6034644/) adducts in both protein and lipid form. The knockout of MPO diminishes CEP generation in skin (PMC6034644), thereby demonstrating the causative relationship between CEP and MPO.

**Author response image 2. sa3fig2:** Additional immunostaining of mouse skin for Keratin 17 (purple), CEP (green) and MPO (red). Similar staining is in S.Fig.5A and quantification is in S.Fig.5B.

Also, the above-mentioned manuscripts show that CEP effects are milder but overall comparable with canonical TLR2 agonists, PAM3SCK4. As we mention in the present manuscript, normal young mice’s tissues are devoid of CEP (which is generated in response to inflammation) with an exception of hair follicles. This is likely attributed to the secretion of MPO by hair follicles (PMID: 36402231) especially in conditions of inflammation (PMID: 32893875). Supplementary Fig.5A,B show that MPO is present at the high level in sebaceous gland (as a part of anti-microbial mechanism). Again, MPO is a secreted enzyme and it is likely to be a source of continuous DHA oxidation into CEP in hair follicles. We also document that both,TLR2 and CEP levels in hair follicles (but not in other tissues-an important point for CEP) are reduced in aging. Likewise, SFig.5A,B shows that MPO secretion in hair follicle is reduced by more than 60% in aging mice. Thus, it is likely that reduced MPO levels in aging hair follicle produce less CEP. Together with reduced TLR2 levels, the lack of CEP might contribute to hair loss in aging.

We show that similar to TLR2, CEP in hair follicles operates via a BMP-7 dependent pathway (see Fig.6). We also provide results using canonical bacterial ligand for TLR2, PAM3CSK4 whose effect on HFSCs proliferation is similar to CEP in a TLR2-dependent manner. TLR2 blocking approaches were used (Supp. Fig.4B, C, D, E, Supp. Fig.5D-5F). It remains to be seen whether CEP is required for the normal hair cycling and whether its administration might improve hair loss in aging subjects.

“The impacts of CEP/TLR2 on proliferation of keratinocytes is still weak. How much of this effect is a result of NFkB activation, and how much is simply due to inhibiting BMP signaling?

Impact of TLR2 on proliferation was demonstrated using a variety of mouse models, from global TLR2 KO to bone marrow chimeras to HFSCs-specific TLR2 KO, again using multiple approaches. The same applies to the effects of CEP as well as to canonical TLR2 ligand, PAM3CSK4, which were demonstrated both in vivo and in culture to be TLR2-dependent (Fig.6MO and Supplementary Fig.4E-D). As for NFkB connection, see our responses above. It seems that the connection between TLR2 and BMP pathway occurs independently of NFkB activation.

1. The links between TLR2 pathway and aging and obesity are only correlative. Although the authors suggest that the reduction of TLR2 expression in aging and obesity may diminish hair growth (Fig 1), there is no direct functional evidence that supports this possibility. If the authors wish to make this claim, they should test the roles of TLR2 and CEP in aging and obesity conditions.”

We show that both, TLR2 and CEP are reduced in aging, and that this pathway contributes to hair cycling and regeneration upon wounding, we do not wish to claim more.

1. More minor points:“Fig.4: The Noggin treatment in TLR2 KO mice is an important experiment. However, it is unclear why Noggin only enhances proliferation (Ki67 level) in HG but not in the bulge. This discrepancy should be addressed.”

As we showed in Fig. 3B-3F, TLR2 HFSC-KO mice have prolonged first telogen. Noggin treatment at the first postnatal telogen promotes telogen to anagen transition in TLR2HFSC-KO characterized by the activation of HG cells prior to the bulge cells. According to the literature, the bulge cells remained silent during the late telogen, however, HGs became Ki67- positive and the proliferation of HG cells contributed to the telogen-to-anagen transition.

(https://www.ncbi.nlm.nih.gov/pmc/articles/PMC2668200/,https://www.sciencedirect.com/science/article/pii/S0022202X15404518?via%3Dihub, https://journals.biologists.com/jcs/article/114/19/3419/34892/Hair-follicle-predetermination).

“Fig.5: Does TLR2 cKO slow down wound healing, in addition to affecting pigmentation and the number of hair follicles?”

In our previous publication, we demonstrated that deletion of TLR2 in HFSC does not affect wound healing process. Instead, endothelial TLR2 promotes wound vascularization and healing.

(see Xiong and all. Timely Wound Healing Is Dependent on Endothelial but Not on Hair Follicle Stem Cell Toll-Like Receptor 2 Signaling.// Journal of Investigative Dermatology, Volume 142, Issue 11, November 2022, Pages 3082-3092.e1).

“There is no panel B in Fig.4. There is no image in Fig 4D. Please correct this properly.”

We corrected Fig.4

“Discussion: The constant production of CEP in homeostatic skin and in the absence of inflammation should be further discussed. Additionally, the possible causes of reducing CEP levels during aging should also be further discussed.”

We explained the sources of CEP generation, such as MPO as a one of the key enzyme, above.

The data on MPO levels in hair follicles of young and old mice are presented in Supplementary Fig.5A,B. Since we previously shown that MPO produces CEP from DHA (PMC6034644), the reduction in MPO in aging is likely to contribute to reduced CEP levels.